# The Tudor SND1 protein is an m⁶A RNA reader essential for replication of Kaposi's sarcoma-associated herpesvirus

**Belinda Baquero-Perez[1,2†]\*, Agne Antanaviciute[3], Ivaylo D Yonchev[4,5], Ian M Carr[3], Stuart A Wilson[4,5], Adrian Whitehouse[1,2,6]\***

[1]School of Molecular and Cellular Biology, Faculty of Biological Sciences, Astbury Centre of Structural Molecular Biology, University of Leeds, Leeds, United Kingdom; [2]Astbury Centre of Structural Molecular Biology, University of Leeds, Leeds, United Kingdom; [3]Leeds Institute of Medical Research, School of Medicine, University of Leeds, St James's University Hospital, Leeds, United Kingdom; [4]Department of Molecular Biology and Biotechnology, University of Sheffield, Sheffield, United Kingdom; [5]Sheffield Institute For Nucleic Acids, University of Sheffield, Sheffield, United Kingdom; [6]Department of Biochemistry and Microbiology, Rhodes University, Grahamstown, South Africa

**\*For correspondence:**
belinda.baquero@upf.edu (BB-P);
A.Whitehouse@leeds.ac.uk (AW)

**Present address:** [†]Virology Unit, Department of Experimental and Health Sciences, Universitat Pompeu Fabra, Barcelona, Spain

**Competing interests:** The authors declare that no competing interests exist.

**Abstract** $N^6$-methyladenosine (m⁶A) is the most abundant internal RNA modification of cellular mRNAs. m⁶A is recognised by YTH domain-containing proteins, which selectively bind to m⁶A-decorated RNAs regulating their turnover and translation. Using an m⁶A-modified hairpin present in the Kaposi's sarcoma associated herpesvirus (KSHV) *ORF50* RNA, we identified seven members from the 'Royal family' as putative m⁶A readers, including SND1. RIP-seq and eCLIP analysis characterised the SND1 binding profile transcriptome-wide, revealing SND1 as an m⁶A reader. We further demonstrate that the m⁶A modification of the *ORF50* RNA is critical for SND1 binding, which in turn stabilises the *ORF50* transcript. Importantly, SND1 depletion leads to inhibition of KSHV early gene expression showing that SND1 is essential for KSHV lytic replication. This work demonstrates that members of the 'Royal family' have m⁶A-reading ability, greatly increasing their epigenetic functions beyond protein methylation.
DOI: https://doi.org/10.7554/eLife.47261.001

## Introduction

$N^6$-methyladenosine (m⁶A) is the most prevalent internal modification of eukaryotic messenger RNAs (mRNAs). Despite the identification of this modification over four decades ago, only recently have major breakthroughs in our understanding of this modification been made due to the development of m⁶A-seq, which allows immunoprecipitation of fragmented m⁶A-modified RNAs followed by next-generation sequencing (NGS) analysis. m⁶A-seq and subsequent enhanced versions of the technique led to transcriptome-wide maps of cellular m⁶A modification (*Dominissini et al., 2012*; *Meyer et al., 2012*; *Patil et al., 2016*). m⁶A writers have also been identified, a catalytically active methyl-transferase, METTL3 (*Wang et al., 2016*), together with adapter components, catalyses m⁶A addition onto specific RNA sequences containing the DRm⁶ACH motif, where D = A, G or U; R = A or G; H = A, C or U. The RNA demethylases FTO (*Jia et al., 2011*) and ALKBH5 (*Zheng et al., 2013*) can erase m⁶A marks offering a dynamic regulation of m⁶A status in RNA. Importantly, a family of effector m⁶A readers have also been identified comprising five YT521-B homology (YTH) domain-containing proteins. YTH readers directly bind m⁶A in target RNAs through an aromatic cage

**eLife digest** When a cell needs to make a protein, it reads from the master copy of the gene in the DNA and prints out temporary duplicates called mRNA. These duplicates then act as templates for protein production. Both DNA and mRNA can be further modified by adding on chemical tags that recruit specific proteins. While chemical modifications in DNA are known to control the activity of genes, their role in mRNA is only just being uncovered.

One of the most common chemical modifications in mRNA is the addition of a methyl group called m⁶A. This methyl group has also been found in the mRNA of certain viruses, including the Kaposi's sarcoma-associated herpesvirus (KSHV) which causes cancer. Recent work has shown that a family of proteins, known as the YTH family, can recognise and bind to this specific methyl group and regulate the rate at which mRNA degrades. To investigate whether other proteins can recognise m⁶A, Baquero-Peŕez et al. mapped the m⁶A residues of mRNAs encoded by KSHV genes and looked at which proteins the methyl mark interacts with.

The experiments revealed that a family of proteins – nicknamed the 'Royal family' – that recognise methyl groups in proteins, can also bind to mRNA that contains m⁶A. Baquero-Peŕez et al. showed that a member of this family, SND1, can read the m⁶A methyl mark on mRNAs from both the virus and the host cell. Further experiments showed that SND1 binds to and stabilises a viral mRNA which provides the template for a protein that the virus needs to replicate. When SND1 was removed from human immune cells infected with KSHV, this caused the virus to replicate less efficiently.

The discovery that the Royal family of proteins can recognise methylated mRNA as well as methylated proteins suggests that there may be a common feature that allows proteins to read methylation. Understanding the shape of this feature could lead to new treatments that block viruses from making the proteins they need to replicate.

DOI: https://doi.org/10.7554/eLife.47261.002

($Liao et al., 2018$), directing their targets towards different biological fates. YTHDF2 recruits the CCR4-NOT deadenylase complex and promotes degradation of m⁶A-containing RNAs (*Du et al., 2016*; *Wang et al., 2014*) while YTHDF1, YTHDF3 and YTHDC2 stimulate mRNA translation of their targets (*Wang et al., 2015*; *Shi et al., 2017*; *Hsu et al., 2017*) and YTHDC1 regulates mRNA splicing (*Xiao et al., 2016*). Whether other proteins can directly recognise m⁶A has remained a question to date. Other RNA-binding proteins can read m⁶A indirectly, in a so-called 'm⁶A switch' mechanism, in which m⁶A destabilises the local RNA structure in hairpins allowing recruitment of either hnRNPC to U-tracts (*Liu et al., 2015*) or hnRNPG to a purine-rich motif (*Liu et al., 2017*), both readers then regulate alternative splicing. Recently, IGF2BP proteins were reported to enhance mRNA stability and translation of m⁶A-containing RNAs and proposed to be a novel class of m⁶A readers (*Huang et al., 2018*).

Due to recent transcriptome-wide m⁶A mapping of multiple viruses (*Kennedy et al., 2016*; *Lichinchi et al., 2016a*; *Tirumuru et al., 2016*; *Gokhale et al., 2016*; *Lichinchi et al., 2016b*; *Courtney et al., 2017*; *Tsai et al., 2018*; *Hesser et al., 2018*; *Tan et al., 2018*), it is becoming evident that there is an interplay between m⁶A-decorated viral RNA and cellular m⁶A machinery, resulting in modulation of viral replication output.

Kaposi's sarcoma-associated herpesvirus (KSHV) is a DNA virus responsible for several Acquired Immuno Deficiency Syndrome-associated aggressive malignancies (*Baquero-Pérez and Whitehouse, 2015*; *Schumann et al., 2017*). KSHV has a biphasic life cycle, with a latent phase and a productive lytic phase. During latency, the KSHV episome is dormant in the nucleus of endothelial progenitor cells and B-lymphocytes (*Giffin and Damania, 2014*), with few viral latent genes expressed. Under various stimuli, the KSHV episome can be reactivated into lytic replication, leading to the ordered expression of more than 80 viral proteins. The replication and transcription activator (RTA) viral protein, which is encoded from open reading frame 50 (ORF50), is the first lytic protein produced and is essential for the latent-lytic switch (*Guito and Lukac, 2012*).

Here we accurately decipher the KSHV m⁶A epitranscriptome using a novel m⁶A peak-calling algorithm and characterise m⁶A readers that specifically interact with KSHV m⁶A-modified RNA

sequences found in *ORF50* and *ORF37*. Through the use of RNA affinity, in addition to YTH readers, eight members from the Tudor domain 'Royal family', including SND1 (Staphylococcal nuclease domain-containing protein 1), were specifically enriched in m6A-modified KSHV sequences. The 'Royal family' is a well-characterised family that reads methylated residues in histones through the use of an aromatic cage (*Chen et al., 2011*) which is structurally similar to the one found in YTH readers (*Luo and Tong, 2014*). Electromobility shift assays (EMSAs) demonstrate the ability of specific Royal domains to selectively bind m6A-modified RNA. As these domains do not bind all m6A-modified RNA sequences it suggests this binding may occur in a RNA secondary structure/sequence-dependent manner. We further developed a modified RIP-seq technique, which significantly improves the resolution of standard RIP, accompanied by a unique bioinformatic analysis to characterise for the first time the transcriptome-wide binding profile of SND1 to cellular and KSHV mRNAs, revealing SND1 as a *bona fide* RNA-binding protein that targets m6A-modified RNAs in KSHV-infected cells, including the extensively m6A-modified *ORF50* RNA. SND1 eCLIP (enhanced crosslinking immunoprecipitation) analysis using publically available datasets deposited in the ENCyclopedia Of DNA Elements (ENCODE) further confirmed that SND1 has a binding profile similar to other m6A reader proteins. Importantly, depletion of SND1 in KSHV-infected cells significantly reduced the stability of unspliced *ORF50* RNA and led to markedly reduced levels of RTA protein together with a global impairment of KSHV lytic replication. Furthermore, we show that m6A-modification in *ORF50* RNA regulates SND1 binding to this RNA, particularly to the unspliced form. These data identify SND1 as an essential m6A reader for KSHV lytic replication and implicate the 'Royal family' as a family which comprises m6A readers. This, considerably expands the landscape of m6A readers and the epigenetic functions of Royal members beyond protein methylation.

## Results

### The KSHV transcriptome is extensively m6A-methylated in a cell type-specific manner

We have previously developed dedicated software (m6aViewer) which implements a novel m6A peak-calling algorithm that identifies high-confidence methylated residues with more precision than previously described approaches (*Antanaviciute et al., 2017*). Utilising this software we mapped m6A modifications in the KSHV transcriptome by performing m6A-seq in TREx BCBL1-Rta cells, a BCBL1-based, primary effusion lymphoma B-cell line containing latent KSHV episomes capable of doxycycline-inducible reactivation of lytic replication. We carried out m6A-seq in latent cells and cells undergoing lytic replication for 8 hr and 20 hr post-induction in two biological replicates. In latent cells, we consistently observed m6A peaks in six viral RNAs, including *ORF72* and *ORF73*. At 8 hr post-reactivation, 33 m6A peaks were identified in 21 KSHV ORFs in both biological replicates (*Figure 1—figure supplement 1a*). At 20 hr post-reactivation, 75 m6A peaks were mapped in 42 KSHV ORFs (*Figure 1a*). The positions of these m6A peaks were highly reproducible across the different time points and replicates (*Figure 1—figure supplement 1b*). 12 viral m6A peaks were further validated by two-step quantitative reverse transcription PCR (qRT-PCR) (*Figure 1—figure supplement 2*). m6A peaks in cellular RNAs were also consistently called, with 18,946 m6A peaks common to both replicates in latent cells and 18,935 and 13,926 m6A peaks in cells reactivated for 8 hr and 20 hr, respectively (*Figure 1b*, *Supplementary file 4*). Cellular m6A peaks remained mostly unchanged between latent and 8 hr of lytic replication. Lower detection of m6A peaks at 20 hr post-reactivation was observed in part due to KSHV-mediated host cell shut-off (*Conrad, 2009*) (*Figure 1c*). However, the majority of uniquely identified m6A peaks at this time point were found in RNAs whose expression was increased during lytic replication (*Figure 1c*). Cellular m6A peaks were enriched in the coding region (CDS) and 3' untranslated region (UTR) throughout KSHV infection (*Figure 1d*). In viral mRNAs, the majority of m6A peaks were located in the CDS (*Figure 1e*). We also confirmed the DRm6ACH motif both in cellular and viral mRNAs (*Figure 1f and g*, respectively). Approximately 60% of viral m6A peaks contained the GGm6AC[G/U] motif, a significant over-representation of the motif when compared to the expected DRm6ACH frequency.

A further comparison using our dedicated software was performed between our m6A-seq datasets and previously published m6A-seq studies carried out in TREx BCBL1-Rta cells (*Tan et al., 2018*) and in a renal carcinoma cell line infected by recombinant KSHV BAC16 (iSLK cells) (*Hesser et al.,*

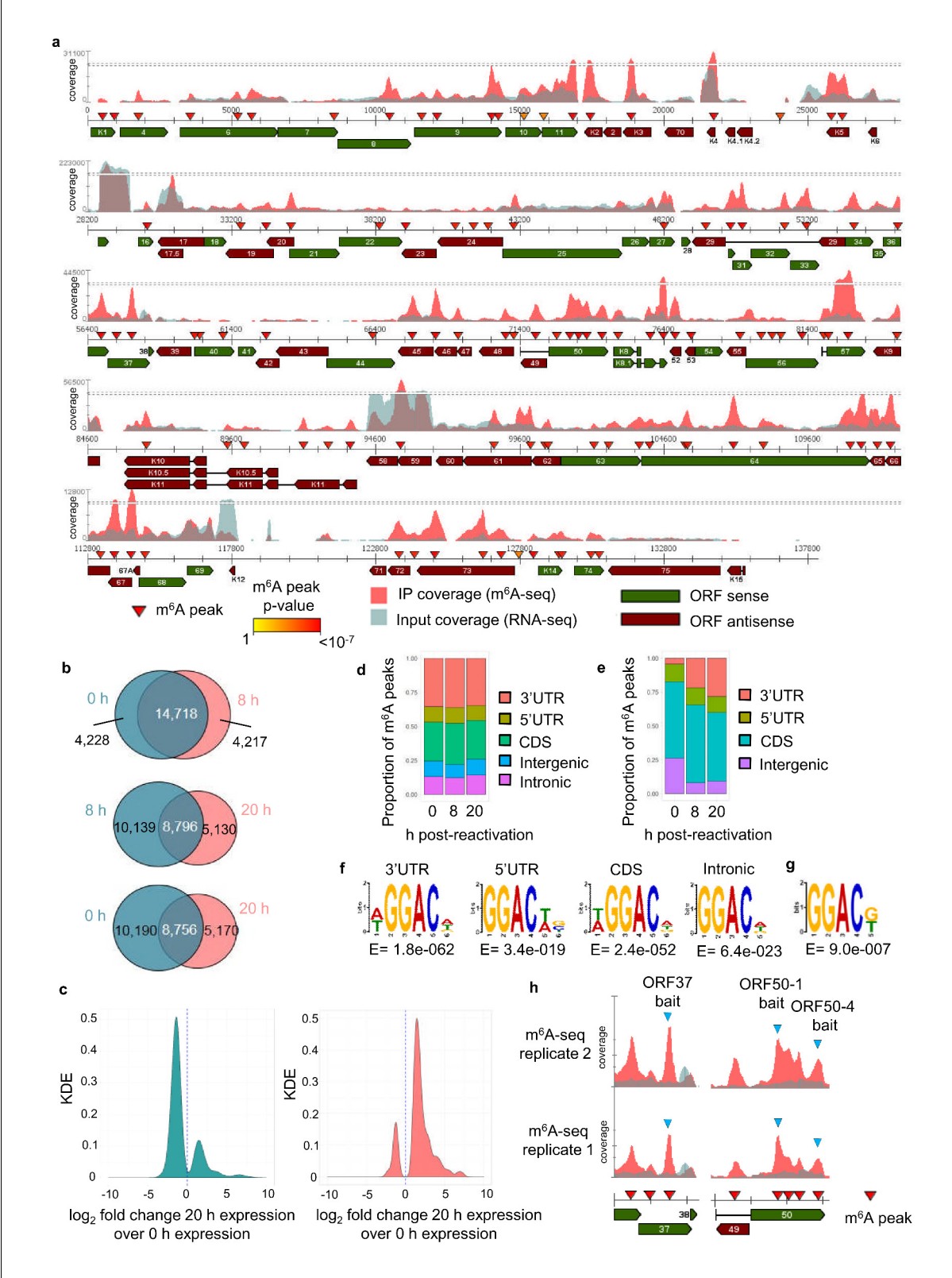

**Figure 1.** The KSHV transcriptome is extensively m⁶A-modified. (a) KSHV epitranscriptome map in TREx BCBL1-Rta cells at 20 hr post-lytic reactivation. m⁶A-IP reads and input reads are shown. m⁶A peaks identified by m6AViewer software with associated p-values are indicated. (b) Number of cellular m⁶A peaks consistently called by m6AViewer software in two biological replicates throughout KSHV infection. (a,b) To define significantly enriched m⁶A peaks in both viral and cellular RNAs, a minimum fold change of m⁶A-IP reads over input reads of $\geq$ 1.5 and a FDR < 5% was required in both

*Figure 1 continued on next page*

## eLIFE Research article

Microbiology and Infectious Disease

*Figure 1 continued*

biological replicates. Peaks positions were considered overlapping between replicates if the calls were within 100 nucleotides between corresponding positions. (c) Kernel density estimate (KDE) on the RNAs coding for the 10,190 m⁶A peaks present in latent but not 20 hr reactivated cells (left graph) and on the RNAs coding for the 5,170 m⁶A peaks present at 20 hr reactivation but not in latent cells (right graph). (d) Distribution of m⁶A peaks in five topological regions of cellular RNAs during latent and lytic KSHV replication. UTR, untranslated region. CDS, coding region. (e) Distribution of m⁶A peaks in four topological regions of viral RNAs during latent and lytic KSHV replication. UTR, untranslated region. CDS, coding region. (f) Most significantly enriched DRm⁶ACH consensus identified in m⁶A peaks present in four topological regions of cellular transcripts. MEME software was used for motif analysis. (g) Most significantly enriched DRm⁶ACH motif found in viral transcripts. MEME software was used for motif analysis. (h) RNA affinity baits were centred on the closest GGACU motif to the second m⁶A peak of open reading frame (ORF37) and the first and fourth m⁶A peaks of the second exon of *ORF50* transcript (blue triangles). m⁶A-IP reads and input reads for *ORF37* and *ORF50* transcript from two biological replicates at 20 hr post-viral reactivation are shown.

DOI: https://doi.org/10.7554/eLife.47261.003

The following figure supplements are available for figure 1:

**Figure supplement 1.** The KSHV transcriptome is differentially m⁶A-modified during the course of reactivation.
DOI: https://doi.org/10.7554/eLife.47261.004

**Figure supplement 2.** Validation of m⁶A peaks.
DOI: https://doi.org/10.7554/eLife.47261.005

*2018*; *Tan et al., 2018*). All raw sequencing data were subjected to quality control and processing as described in methods and m⁶A peaks were identified with m6aViewer. Peaks were filtered to keep only those with a minimum of 20 mean reads, 1% FDR (benjamini-hochberg) and an enrichment of 4-fold m⁶A-IP/input. After applying these cut-offs, our TREx BCBL1-Rta cells datasets contained twice as many m⁶A peaks as compared to the Tan dataset in the same cell line. ~17,000 and~10,000 cellular m⁶A peaks were identified in our dataset during latency and lytic replication, respectively, while the Tan dataset yielded ~9,000 m⁶A peaks during latency and ~4,500 m⁶A peaks during lytic replication (*Figure 2a*). The most cellular m⁶A peaks were identified in the Hesser datasets in iSLK cells with ~25,000 m⁶A peaks. A direct one-to-one reciprocal overlap comparison between each of our m⁶A peaks and those found in the other datasets was not possible, mostly due to vastly different peak widths (with the other datasets having narrower peaks and containing short read length coupled to single end data). In such situations, many peaks cannot be mapped one-to-one, with multiple peaks often overlapping a single one when comparing datasets, which leads to overlap quantifications that are not easily interpretable. Therefore, we compared the overlap of m⁶A-modified transcripts identified between all three studies. In TREx BCBL1-Rta cells, 72% of our 5,830 m⁶A-modified transcripts in latent cells were also present in the same cell line during latency in the Tan dataset, while 52% of our 4,059 m⁶A-modified transcripts in lytic cells were common to the lytic Tan dataset (*Figure 2b*).~80% of m⁶A-modified transcripts in our TREx BCBL1-Rta dataset were also identified in the Hesser dataset which mapped m⁶A in iSLK cells (*Figure 2b*). This analysis shows a high reproducibility identifying m⁶A-modified transcripts in KSHV-infected cell lines by three different groups.

Regarding viral m⁶A peaks, again, after applying the same data processing and cut-offs, we identified almost twice as many m⁶A peaks during lytic replication than the Tan dataset (*Figure 2a*). We then compared the m⁶A-IP coverage maps of the KSHV transcriptome in lytic TREx BCBL1-Rta cell lines, overall, viral m⁶A peaks were strikingly similar in both studies (*Figure 2c*). Of note, m⁶A peaks in *ORF50* RNA were consistently mapped in similar regions in both iSLK and TREx BCBL1-Rta cell lines (*Figure 2—figure supplement 1*), with the highest similarity found between TREx BCBL1-Rta cell lines. Although, in particular instances, m⁶A peaks differed between studies, for example *ORF56* and *ORF64* had a distinct m⁶A peak in this study but not in Tan et al. (*Figure 2c and d*, purple boxes), while K12 was m⁶A-modified in Tan et al. but not in our study (*Figure 2c and d*, green box). Intriguingly, when we compared the m⁶A-IP coverage maps between our lytic TREx BCBL1-Rta cells and lytic iSLK cells from Tan et al., although clear common peaks were present, these maps differed more than the TREx BCBL1-Rta cells (*Figure 2c and d*, blue lines) suggesting that the KSHV transcriptome may be differentially modified depending on the cell type, suggesting potential epitranscriptomic remodelling of silencing and activating m⁶A motifs may take place differently between cell types (*Hesser et al., 2018*; *Tan et al., 2018*).

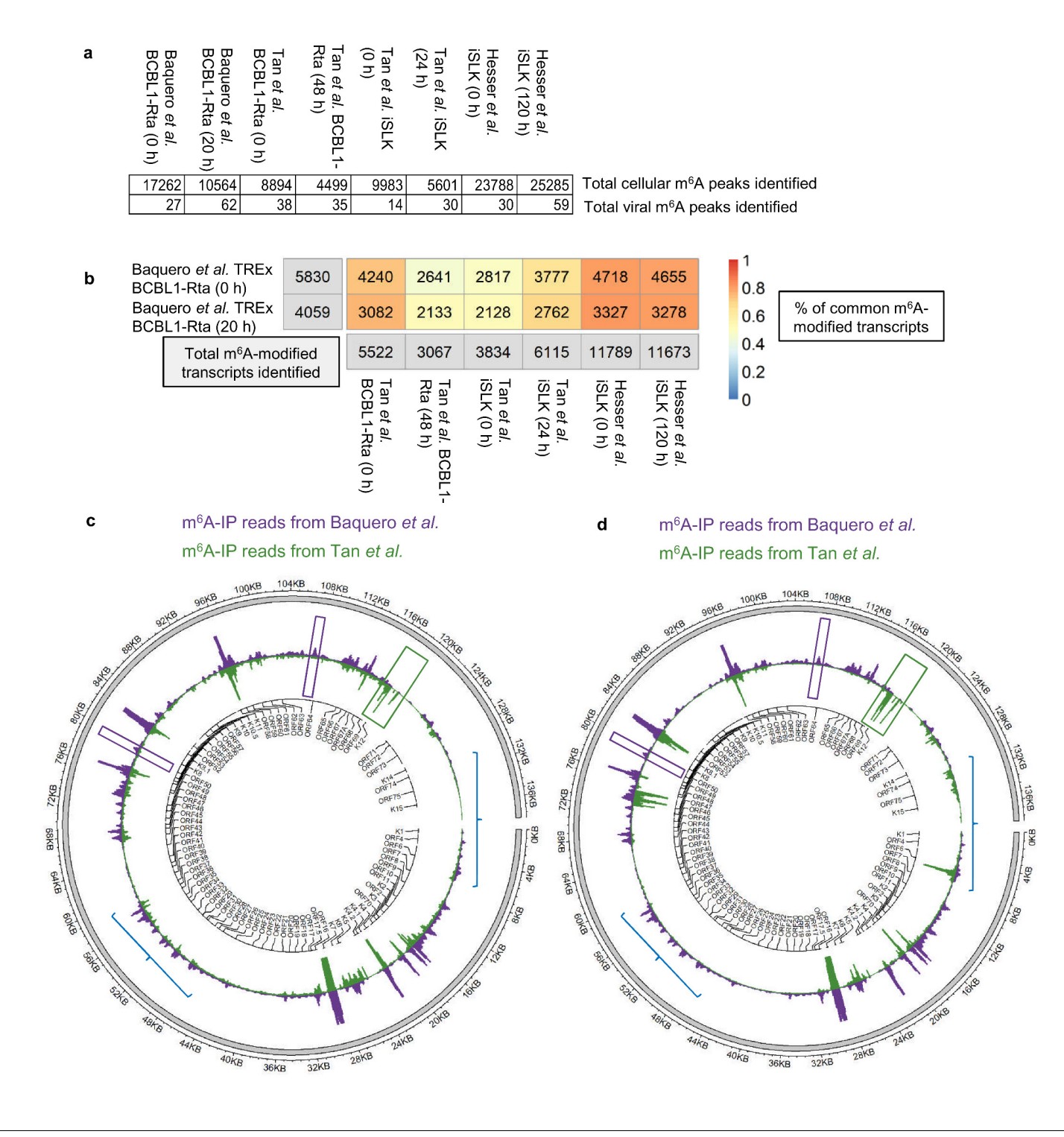

**Figure 2.** Comparison of epitranscriptomic maps in different KSHV-infected cell lines (**a, b**) Fastq files were downloaded from SRA (GEO accessions: GSE93676 and GSE104621) using sratoolkit. Raw sequencing data were subjected to quality control and processing as described in methods. Reads were aligned to a combined human hg38 and KSHV genome reference. m6A peaks were identified with m6aViewer version 1.6.1. Peaks were filtered to keep only those with a minimum of 20 mean reads, 1% FDR (benjamini-hochberg) and an enrichment of 4-fold m6A-IP/input. Overlapping m6A peaks between replicates were collapsed using GenomicRanges R package and only peaks detected across all replicates were kept for further comparisons. (**b**) Total number of overlapping m6A-modified transcripts identified between replicates are displayed for each dataset and time point in the grey boxes. The percentage of the total m6A-modified transcripts identified in our dataset that overlap with previously published studies in TREx BCBL1-Rta

*Figure 2 continued on next page*

*Figure 2 continued*

and iSLK cells (**Hesser et al., 2018**; **Tan et al., 2018**) is shown, with 1 being 100%. (**c**) Comparison of the lytic epitranscriptomic map from our study and the one performed by Tan et al., both carried out in TREx BCBL1-Rta. Circos plot of m⁶A-IP read coverage across the viral episome is shown for both studies. Coverage tracks were scaled to account for differences in library sizes. KSHV gene positions are indicated by the central track. Some differences in the identified m⁶A peaks between the two datasets are highlighted. (**d**) As described in c, but comparing Baquero et al. lytic TREx BCBL1-Rta cells with lytic iSLK cells from Tan et al.

DOI: https://doi.org/10.7554/eLife.47261.006

The following figure supplement is available for figure 2:

**Figure supplement 1.** Viral *ORF50* RNA is consistently detected as m⁶A-modified in different m⁶A-seq datasets.

DOI: https://doi.org/10.7554/eLife.47261.007

## RNA affinity identifies putative m⁶A readers which belong to the Tudor domain 'Royal family'

We were intrigued to determine whether any m⁶A readers uniquely interact with methylated viral mRNAs. Of particular interest were the m⁶A peaks found in the second exon of *ORF50* RNA, as this RNA encodes the latent to lytic switch RTA protein. To this end, RNA affinity coupled to mass spectrometry analysis was performed. Viral RNA baits were centred on the closest GGACU motif to the m⁶A peak positions of the largest peak of *ORF37* and the first and fourth m⁶A peaks of the second exon of *ORF50* (**Figure 1h**) (hereafter referred to as ORF37, ORF50-1 and ORF50-4 baits respectively). See **Figure 3—figure supplement 1** for m⁶A peak positions from all m⁶A-seq time points. Mass spectrometry analysis revealed that all three m⁶A baits enriched for YTH readers (**Figure 3a**). An intriguing observation was that of the three viral baits, exclusively the methylated ORF50-1 (m⁶A-ORF50-1) distinctly enriched SND1, a Tudor domain-containing protein. SND1 was in the top thirteen enriched protein hits, together with YTHDF1, YTHDF2 and YTHDF3. Further binding validation of YTHDF1 and SND1 in RNA affinity experiments demonstrated that YTHDF1 was greatly enriched in all three methylated baits while SND1 showed a modest but clear enrichment exclusively in m⁶A-ORF50-1 bait (**Figure 3—figure supplement 2**). SND1 binds symmetrically dimethylated arginines (sDMA) via its Tudor domain (**Liu et al., 2010**), thus it harbours the ability to selectively recognise methyl groups. In addition to SND1, m⁶A-ORF50-1 bait also prominently pulled down three spliceosomal proteins known to interact with the Tudor domain of SND1, snRNP200, snRNP116 and PRPF8 (**Yang et al., 2007**). These proteins were within the top fifteen enriched protein hits. Multiple proteins related to RNA processing were also enriched in this bait (**Supplementary file 1**). SND1 belongs to the Tudor domain 'Royal family', comprising five subfamilies: Tudor, plant agenet, chromo, PWWP and MBT (**Maurer-Stroh et al., 2003**). Members from each subfamily share a structurally related β-barrel that harbours an aromatic cage implicated in binding methylated arginines and lysines (**Chen et al., 2011**). Intriguingly, the aromatic cage used for m⁶A recognition by YTH readers is structurally similar to the one present in the 'Royal family' (**Luo and Tong, 2014**; **Li et al., 2014**; **Xu et al., 2015**; **Xu et al., 2014**). Thus, further scrutiny for Royal members was carried out in the mass spectrometry data. Strikingly, several Royal members were also enriched in methylated viral baits. All three plant agenet members, which were recently identified as RNA sequence-dependent m⁶A readers (**Edupuganti et al., 2017**; **Arguello et al., 2017**), were exclusively bound to m⁶A-ORF50-1 bait (**Figure 3a**). Three PWWP domain-containing proteins were also enriched in m⁶A-ORF50-1 bait while m⁶A-ORF37 bait retrieved the chromo protein CBX3 (**Figure 3a**). These results suggest that Royal domains may be capable of reading methyl-decorations not only in proteins but also in RNA. None of the indirect m⁶A readers, hnRNPA2B1, hnRNPC or hnRNPG, or IGF2BP proteins were enriched in any of the baits (**Supplementary file 2**). Eight proteins with methyl-transferase activity were also recruited to methylated baits (**Supplementary file 2**), of these, METTL16 is the second m⁶A methyltransferase identified to date (**Pendleton et al., 2017**), which methylates structured RNAs where the adenosine is present in a bulge in the nonamer sequence UAC<u>A</u>GAGAA, where <u>A</u> is modified (**Mendel et al., 2018**). The remaining identified proteins may play a previously uncharacterised role in the m⁶A RNA metabolism. Taken together, these results identify several members from the 'Royal family' as putative m⁶A readers.

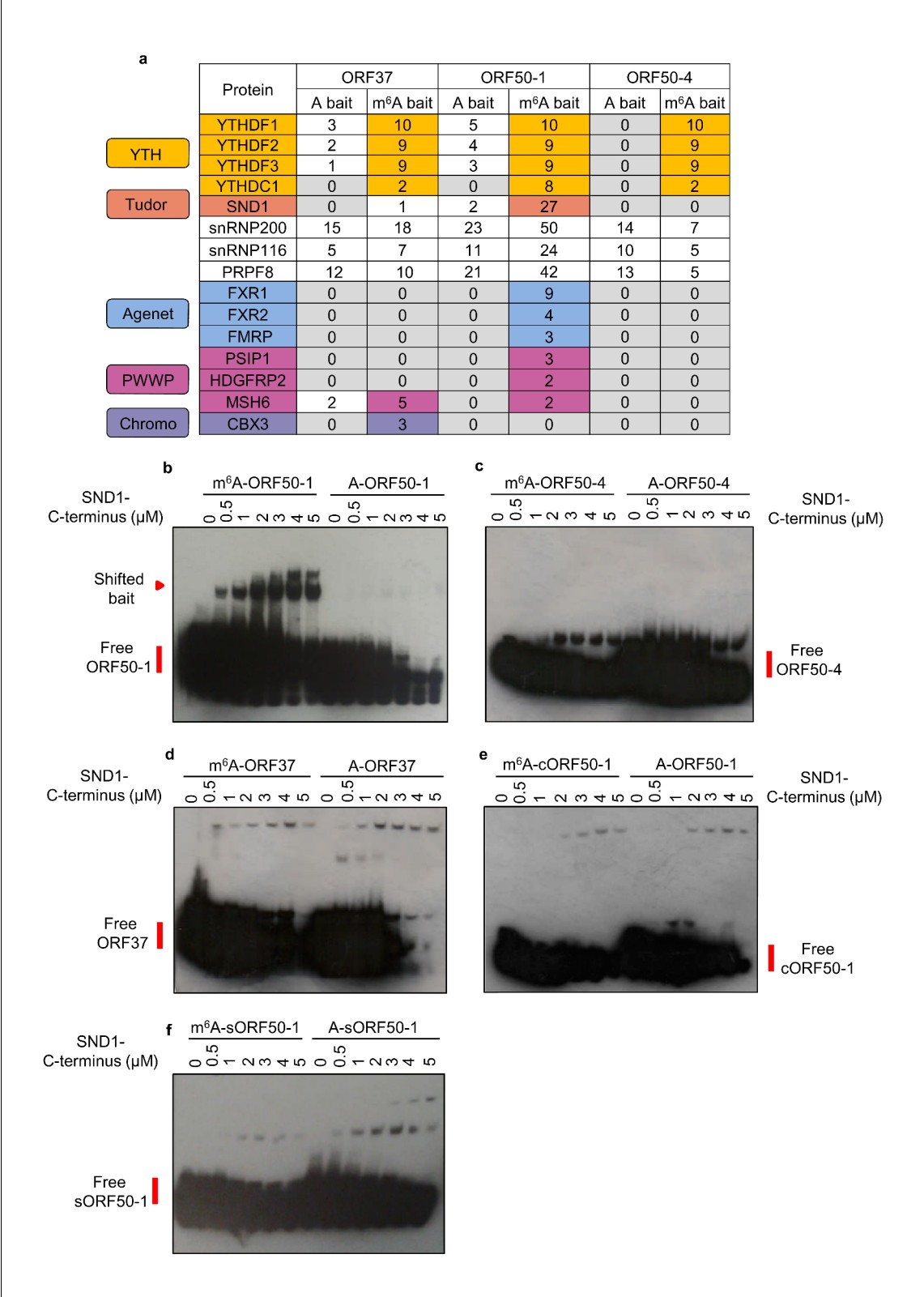

**Figure 3.** LC-MS/MS identifies members from the 'Royal family' as putative m⁶A readers. (a) Mass spectrometry results of RNA affinity pull-downs using m⁶A-modified and unmodified (control) biotinylated viral RNA baits. A single RNA affinity biological replicate for each RNA bait was carried out. Statistical significance of enrichment was not determined, instead, all proteins identified for a given methylated and control bait were sorted by total number of peptide spectrum matches (PSM's) for the m⁶A bait. Proteins were then considered as enriched in methylated baits if the number of PSM's

*Figure 3 continued on next page*

*Figure 3 continued*

assigned to the protein was at least double in the methylated bait compared with the control bait. In this table, the number of unique peptides assigned to each protein is shown for each bait for clarity. (**b–f**) EMSAs were carried out using recombinant SND1-C-terminus protein (residues 548–910) and biotinylated viral RNA baits. A cropped version of ORF50-1 bait (cORF50-1) and a stable version of ORF50-1 bait (named sORF50-1) in which the beginning of the stem is made to have strong base pairing was also tested. Representative EMSAs from six and two independent protein purifications for b and c-f, respectively.

DOI: https://doi.org/10.7554/eLife.47261.008

The following figure supplements are available for figure 3:

**Figure supplement 1.** m$^6$A peak positions from all m$^6$A-seq biological replicates studied.

DOI: https://doi.org/10.7554/eLife.47261.009

**Figure supplement 2.** Validation of RNA affinity pull-downs.

DOI: https://doi.org/10.7554/eLife.47261.010

**Figure supplement 3.** RNA secondary structure prediction of viral RNA baits.

DOI: https://doi.org/10.7554/eLife.47261.011

**Figure supplement 4.** A fraction of free m$^6$A-ORF50-1 bait migrates slower than A-ORF50-1 bait.

DOI: https://doi.org/10.7554/eLife.47261.012

**Figure supplement 5.** Representative coomassie-stained SDS-PAGE gels for all recombinant proteins used.

DOI: https://doi.org/10.7554/eLife.47261.013

**Figure supplement 6.** Recombinant proteins containing the PWWP domain of PSIP1 and the plant agenet domain of FXR1 selectively bind m$^6$A-modified hairpins.

DOI: https://doi.org/10.7554/eLife.47261.014

**Figure supplement 7.** Neither CBX3 chromodomain nor GST protein show selectively for binding m$^6$A-modified RNA.

DOI: https://doi.org/10.7554/eLife.47261.015

## Royal domains display selectivity for specific m$^6$A-modified RNA hairpins

We next determined whether the Royal domains from the Royal members enriched in methylated viral RNAs display affinity for m$^6$A-modified RNA in the absence of any other protein interaction. The complete Royal Tudor domain of SND1 (residues 650–910) is required for sDMA-binding (*Liu et al., 2010*), consequently, a GST-recombinant protein containing these residues (548-910), referred here as SND1-C-terminus, was used in native electromobility shift assays (EMSAs). Notably, SND1-C-terminus shifted m$^6$A-ORF50-1 bait in contrast to the control bait (*Figure 3b*). Neither ORF50-4 nor ORF37 methylated baits were selectively bound by SND1-C-terminus (*Figure 3c and d*, respectively). Note that the membranes had to be overexposed to obtain a good shift signal due to the small amount of shifted RNA in comparison with the free bait, consistent with the modest enrichment of SND1 in m$^6$A-ORF50-1 bait (*Figure 3—figure supplement 2*). Similarly, a weak shift has also been previously observed in EMSAs for FMRP protein (*Edens et al., 2019*) and IGF2BP proteins (*Huang et al., 2018*). RNA secondary structure prediction of all baits indicates that they form strongly base-paired hairpins with an apical loop in which m$^6$A is exposed and an additional large unpaired bulge; however, ORF50-4 and ORF37 feature this unpaired bulge in the middle of their stem (*Figure 3—figure supplement 3*). This may imply that SND1 cannot bind when the unpaired bulge is present at this position irrespective of m$^6$A. In contrast, the beginning of the stem of ORF50-1 shows weak base-pairing with four unpaired bases (*Figure 3—figure supplement 3*, black box) that may allow opening of the hairpin and selective SND1-binding when m$^6$A is present. Curiously, when running m$^6$A-ORF50-1 bait on its own (without any protein), two distinct bands were visualised, one lower band which migrated at the same speed of A-ORF50-1 bait and another higher band migrating slower (*Figure 3—figure supplement 4*). Electrospray ionisation (ESI) of the ORF50-1 baits showed that there were no truncated forms and that the correct molecular weight of a methyl group had been added (15 daltons), demonstrating the purity of the baits (*Figure 3—figure supplement 4*). As m$^6$A can destabilise local RNA structure in hairpins (*Liu et al., 2015*), it seems plausible that m$^6$A addition to ORF50-1 destabilises the hairpin which then migrates slower compared with the non-methylated bait. To test this hypothesis, a cropped version of ORF50-1 bait (cORF50-1) with strong base-pairing throughout the stem (*Figure 3—figure supplement 3*) was tested in EMSAs. Remarkably, SND1-C-terminus did not shift this bait (*Figure 3e*), highlighting that structural features of the RNA ligand are critical for SND1-binding and that the beginning of the m$^6$A-ORF50-1 hairpin

(*Figure 3—figure supplement 3*, boxed region) may be required for an interaction with SND1. We further mutated seven nucleotides to make this region very structured and stable, as demonstrated by the increase in free energy of this stable hairpin (sORF50-1). No specific selectivity for m⁶A-sORF50-1 by SND1-C-terminus was seen (*Figure 3f*), indicating that destabilisation of this region is required for SND1 binding. Shorter exposure of the free m⁶A-sORF50-1 bait did not reveal two distinct bands as the ones present in m⁶A-ORF50-1 hairpin (*Figure 3—figure supplement 4*). To further support the hypothesis that Royal domains harbour selectivity for m⁶A-modified RNA, GST-recombinant proteins containing the Royal domains of FXR1, PSIP1 and CBX3 were tested in EMSAs. Coomassie staining for all recombinant proteins can be seen in *Figure 3—figure supplement 5*. The Royal domains of FXR1 and PSIP1 selectively shifted m⁶A-ORF50-1 bait, in contrast, none of the other baits were bound (*Figure 3—figure supplement 6a and b*, respectively). The CBX3 chromo-domain displayed a very faint shift for m⁶A-ORF50-1 bait, however this shift was not consistent between protein preparations (*Figure 3—figure supplement 7a*) indicating that this domain either may not be able to read m⁶A, or other protein partners could be required for its interaction with m⁶A in vivo. Control glutathione S-transferase (GST), a protein with no RNA-binding properties, was also tested in EMSA showing no selectivity for m⁶A-ORF50-1 bait (*Figure 3—figure supplement 7b*). EMSAs were also performed in the presence of excess herring sperm DNA as a source of non-specific DNA. In the presence of sperm DNA no shift was detected and the free bait remained uncomplexed (*Figure 3—figure supplement 7c*), suggesting that non-specific DNA prevents the interaction between SND1 and m⁶A-ORF50-1 bait. This is not surprising as excess of DNA may sequester SND1 which is a known DNA-binding transcription factor. Together, these results confirm that Royal domains in the absence of any other protein interaction display selectivity for m⁶A-modified RNA, our data also suggests that this selectivity may occur in a RNA secondary structure-dependent manner.

## SND1 is an m⁶A reader in KSHV-infected cells

Next, we aimed to characterise the RNA-binding sites of endogenous SND1 transcriptome-wide during KSHV infection by performing RIP-seq in two biological replicates. Latent and lytic TREx BCBL1-Rta cells that had been reactivated for either 8 or 20 hr were used. We aimed to obtain the best protein binding site resolution by sonicating the majority of the total RNA to <200 bp (*Figure 4—figure supplement 1a and b*). Unexpectedly, after construction of the cDNA library from the SND1-RNA immunoprecipitated (RIP) samples, the majority of the fragments ranged from 150 to 1000 bp (*Figure 4—figure supplement 1c*). Thus, this technique has a binding site resolution of ~1 kB. A transcript-wide analysis enabled us to identify SND1 RNA targets, using a false discovery rate (FDR) < 1% and a fold change of RIP reads over input reads >2, this analysis uncovered SND1 as a *bona fide* RNA-binding protein, yielding 5061 target transcripts (*Supplementary file 5*). Of these, 3319 transcripts were mRNAs and 748 were long non-coding RNAs (lncRNAs). These target RNAs were consistently bound by SND1 during latency and lytic KSHV replication (both at 8 and 20 hr). Gene ontology (GO) analysis revealed that these transcripts code for proteins that are involved in regulating GTPase activity, nervous system development and cell morphogenesis (*Figure 4a*). Next, all highly expressed transcripts identified by RIP-seq (FDR < 1%) were divided into high-confidence targets (those which had at least twice the normalised coverage in RIPs than input) and high-confidence non-targets (those which had at least twice as much coverage in inputs than RIPs) and the proportion of m⁶A-bearing RNAs in each group was determined. Strikingly, we observed that ~50% of high-confidence targets and ~24% of high-confidence non-targets were m⁶A-modified RNAs (*Figure 4b*), representing a marked enrichment of m⁶A-bearing RNAs in the target group. It is worth noting that this transcript-wide analysis will not identify all SND1 targets, as when looking at the SND1-binding profile at the transcript level, it was evident that some RNAs are bound by SND1 at a specific region only (*Figure 4c*).

A positive correlation between the number of m⁶A peaks in a given transcript and the SND1-fold enrichment was not found (*Figure 4d*). These results are not unexpected as they are in agreement with the finding that SND1 does not bind m⁶A indiscriminately. We then mined previously processed PAR-CLIP and m⁶A-seq data sets (*Wang et al., 2014*; *Wang et al., 2015*; *Shi et al., 2017*) from HeLa cells and calculated the percentage of total RNA targets of YTH readers that contain m⁶A-modified transcripts. ~ 65% of all RNA targets contained m⁶A-modified transcripts (*Figure 4—figure supplement 2*). In addition, we compared the overlap of target genes containing or lacking m⁶A

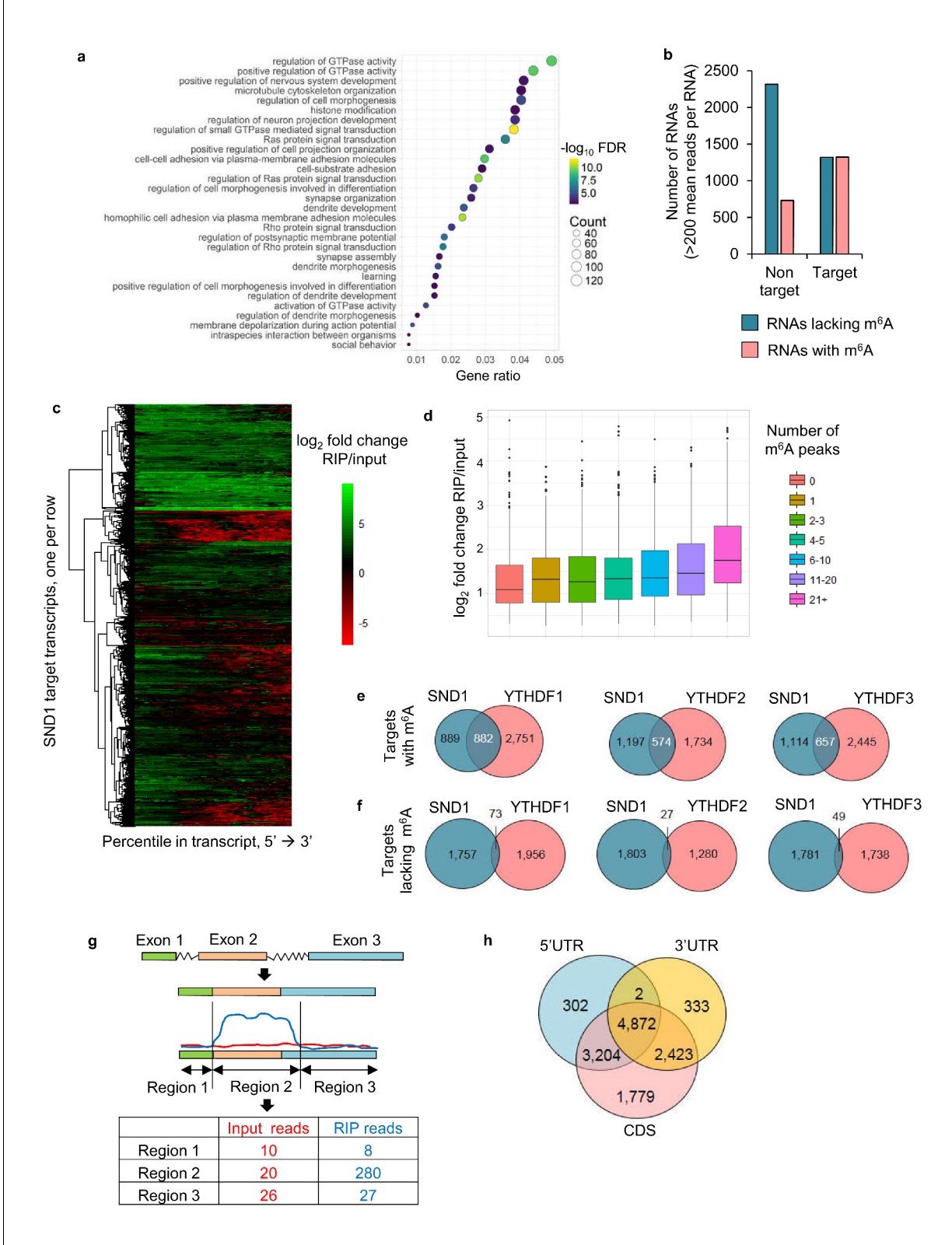

**Figure 4.** SND1 is a *bona fide* RNA-binding protein that targets transcripts bearing m$^6$A modification. (**a**) Most significantly enriched GO terms amongst the 5061 target RNAs (Ensembl transcripts) identified by RIP-seq which are consistently bound by SND1 at all time points (0 hr, 8 hr and 20 hr post-reactivation) in TREx BCBL1-Rta cells. FDR, false discovery rate. (**b**) Consistently bound SND1 RNA targets throughout the course of KSHV infection are enriched in m$^6$A-modified RNAs, while non-targets are depleted. Target transcripts are defined by a fold change RIP/input > 2 while non-targets

*Figure 4 continued on next page*

Figure 4 continued

have a fold change input/RIP > 2. A false discovery rate (FDR) < 1% is applied to RIP-seq data. (c) Hierarchical clustering of fold change RIP/input for SND1 targets. (d) No significant correlation between the number of $m^6A$ peaks in a given RNA and the binding of SND1 as determined by $log_2$ fold change RIP/input. Target transcripts with >300 mean reads of coverage per RNA were used for the analysis. Analysis using a lower expression cut-off showed similar results. (e, f) High-confidence SND1-bound genes (summarised at HGNC gene symbol annotation level) were defined using a cut-off of FDR < 1% and a fold change RIP/input > 2, while $m^6A$ peaks were detected using a FDR < 5% and > 1.5 fold $m^6A$-IP reads over input reads. RNA targets and $m^6A$ peaks for YTH readers were mined from publically available PAR-CLIP and $m^6A$-seq datasets from HeLa cells. (e) Overlap of target genes with $m^6A$ modifications between SND1 and heterologously expressed YTH reader proteins. (f) Overlap of target genes lacking $m^6A$ modifications between SND1 and heterologously expressed YTH reader proteins. (g) For SND1 localised enrichment analysis, introns are collapsed and exons spliced together into a single continuous RNA molecule. Spliced transcripts and introns are then segmented into transcriptomic regions based on changes in their fold change RIP/input. (h) Venn diagram showing the overlap between SND1 enrichment in coding region (CDS) and untranslated regions (UTRs) of SND1 target transcripts identified by localised enrichment analysis.
DOI: https://doi.org/10.7554/eLife.47261.016

The following figure supplements are available for figure 4:

**Figure supplement 1.** Long RNA fragments crosslinked to SND1 are enriched over shorter fragments during RIP.
DOI: https://doi.org/10.7554/eLife.47261.017

**Figure supplement 2.** 65 % of all RNA targets of YTH readers contain $m^6A$-modified transcripts.
DOI: https://doi.org/10.7554/eLife.47261.018

**Figure supplement 3.** Bioinformatic analysis of SND1 RIP-seq data.
DOI: https://doi.org/10.7554/eLife.47261.019

modification between SND1 and each heterologously expressed YTH reader. To our surprise, despite comparing TREx BCBL1-Rta and HeLa cells, we found that ~50%,~32% and~37% of SND1 $m^6A$-modified targets are common to YTHDF1, YTHDF2 and YTHDF3, respectively (*Figure 4e*). When we compared the SND1 targets that lack $m^6A$, only ~4%, ~1% and ~3% were shared with YTHDF1, YTHDF2 and YTHDF3, respectively (*Figure 4f*). These results show that SND1 does not merely co-precipitate with YTH readers and that it has a distinct RNA-binding profile.

To identify SND1 RNA targets that could be missed by transcript-wide analysis, a novel localised enrichment analysis was developed similar to that used in ChIP-seq analysis (see Materials and methods). In brief, both spliced transcripts and introns were segmented into a total of ~750,000 transcriptomic regions based on changes in their fold change RIP/input (*Figure 4g*). Applying a fold change >2 and >50 mean reads per region (FDR < 1%), 32,314 transcriptomic regions were significantly bound by SND1. These regions encompassed 12,915 Ensembl transcripts and after applying a cut-off of >2 fold $m^6A$-IP/input enrichment and a minimum RIP read depth of 50,~40% of these transcripts were $m^6A$-modified. 4872 of these total targets showed SND1-binding throughout 5'UTR, CDS and 3'UTR (*Figure 4h*). We then focused on analysing SND1-enriched intronic regions, as introns tend to be longer than spliced RNAs and RIP-seq has a low resolution, we hypothesised that if SND1 is an $m^6A$ reader we might observe $m^6A$ peaks overlapping with SND1-enriched regions in introns. Out of the total 32,314 SND1-enriched regions, 2563 regions were found in introns. Of the latter, 516 intronic regions (20%), directly overlapped with $m^6A$ peaks (*Figure 5a and e*). As a control for random overlap, 8,328 SND1-unbound intronic regions were used, these were defined using a fold change RIP/input < 0.5 and >50 mean reads per region (FDR < 1%). Of these, only ~1.3% (109) directly overlapped with $m^6A$ peaks (*Figure 4e*). SND1-enriched regions overlapping $m^6A$ peaks in UTRs (*Figure 5b*) and lncRNAs (*Figure 4—figure supplement 3a*) were also observed. In 5'UTRs and 3'UTRs we found ~40% overlap with $m^6A$ peaks in SND1-enriched regions compared to only ~20% overlap of SND1-unbound regions (*Figure 5e*).

Next, we set out to investigate the SND1-enriched intronic regions for enriched motifs. Interestingly, a U-tract was the most significant motif identified (*Figure 4—figure supplement 3b*). In addition, several GAC-containing motifs appeared as significantly enriched (*Figure 4—figure supplement 3c*). When searching for motifs that were 30 nucleotides long in SND1-bound intronic regions containing $m^6A$ peaks, a U-tract immediately followed by an $m^6A$ motif was identified (*Figure 4—figure supplement 3d*). In SND1-bound intronic regions that do not have $m^6A$ peaks, a U-tract was also the most enriched motif. Notably, U-rich motifs found adjacent to $m^6A$ residues are also targeted by RBM15/15B and hnRNPC (*Patil et al., 2016*; *Liu et al., 2015*).

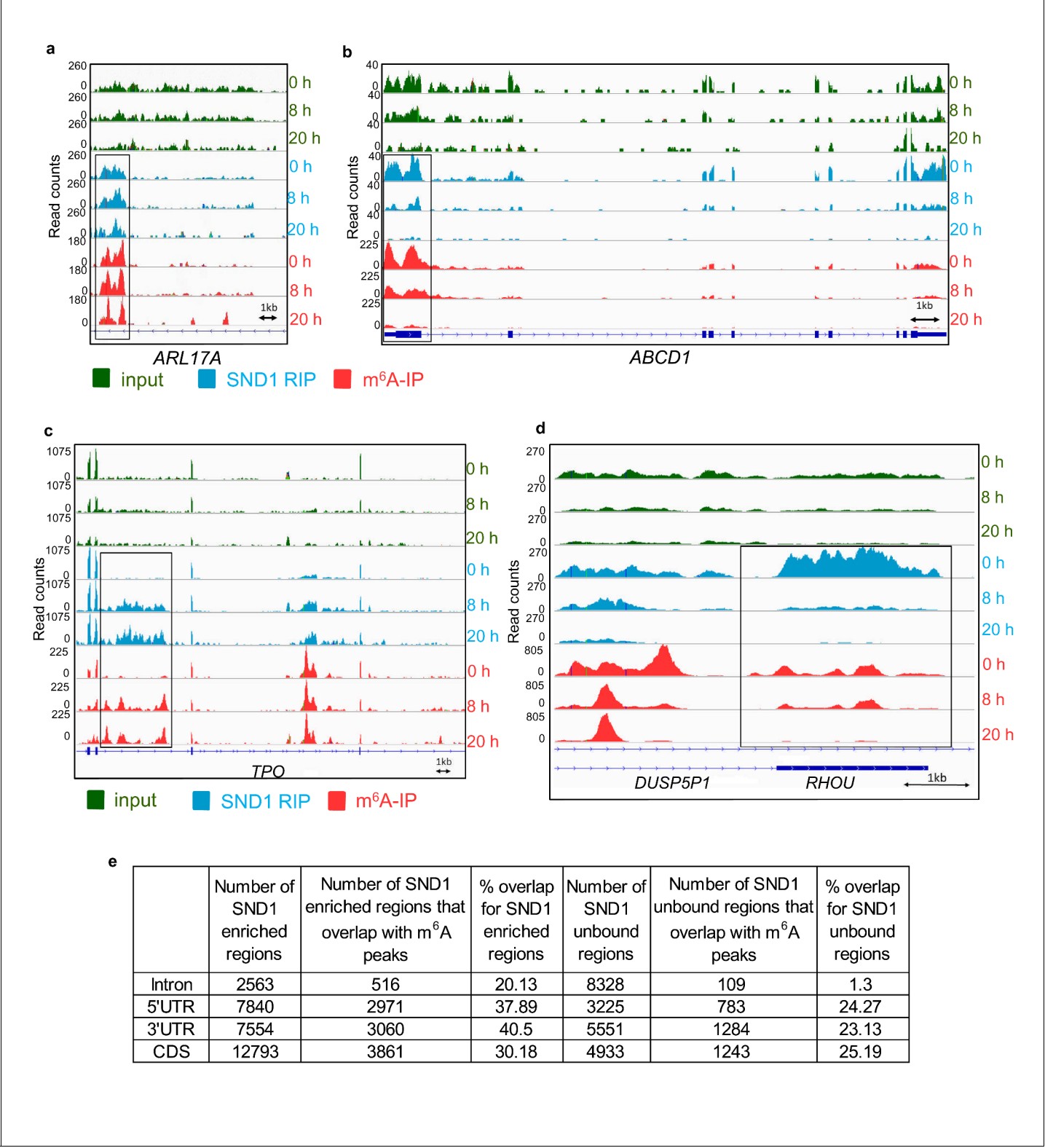

**Figure 5.** SND1 RNA-binding sites specifically overlap m⁶A peaks (a, b) Genome sequencing tracks from latent (0 hr) and lytic (8 hr and 20 hr post-reactivation) TREx BCBL1-Rta cells depicting input (green) and RIP (blue) coverage. m⁶A-IP reads (red) are also shown. Blue boxes indicate exons while blue lines represent introns. SND1 overlaps with m⁶A peaks are evident in introns (*ARL17A*) and 5'UTRs (*ABCD1*). Note that in *ABCD1*, SND1 binding and m⁶A peaks are both reduced at 8 hr post-reactivation while at 20 hr post-reactivation methylation and SND1 binding signal are both lost with decreasing expression in inputs. (c, d) Dynamic SND1 binding to m⁶A-modified regions in SND1 target transcripts during KSHV infection. Genome

*Figure 5 continued on next page*

Figure 5 continued

tracks depicting sequencing read coverage from input (green) and RIP (blue) samples. m$^6$A-IP reads (pink) are also shown. Note that in contrast to *ABCD1*, *RHOU/DUSP5P1* remain highly expressed even at 20 hr post-reactivation, suggesting the coupled loss of methylation/SND1-binding is independent of the ability to detect these events due to loss of expression. (e) SND1 regions consistently enriched across the three time points studied (0 hr, 8 hr and 20 hr) were defined by applying a fold change RIP/input >2 and >50 mean reads per region (FDR < 1%). Consistently SND1-unbound regions throughout KSHV infection were defined using a fold change RIP/input <0.5 and >50 mean reads per region (FDR < 1%). For all m$^6$A overlap analysis, the cut-off for an m$^6$A peak was defined as having 50 read paired at the tallest point in the m$^6$A peak and a 2-fold enrichment of m$^6$A-IP reads over input reads using a FDR < 1%.

DOI: https://doi.org/10.7554/eLife.47261.020

RIP-seq also enabled us to identify differential SND1-binding to target RNAs that correlated to both differentially m$^6$A-modified RNAs and the status of KSHV infection. During latency, one of the introns in *TPO* transcript was not m$^6$A-modified and failed to bind SND1. In contrast, at 8 hr and 20 hr post-reactivation, several adenosines in this intron are m$^6$A-modified and SND1 binding ensues (*Figure 5c*). The reverse scenario was also observed for example in the overlapping DUSP5P1-*RHOU* transcripts, in which specific m$^6$A peaks and SND1-enriched regions present at 0 hr and 8 hr post-reactivation disappeared at 20 hr of lytic reactivation (*Figure 5d*). A list of these differential SND1-binding events to target RNAs can be accessed in *Supplementary file 6*. Taken together, these results demonstrate that SND1 targets m$^6$A-modified regions in KSHV-infected cells at the transcriptome level.

To further address the binding profile for SND1 we re-analysed multiple eCLIP datasets from the ENCODE consortium for established m$^6$A reader proteins and SND1 (*Van Nostrand, 2017*). Since the eCLIP datasets were derived from HepG2 cells, we firstly assessed the overlap between the m$^6$A profile in latent TREx BCBL1-Rta cells and HepG2 cells using an existing m$^6$A-seq dataset from HepG2 cells (*Huang et al., 2018*) (*Figure 6a*). This revealed an extensive overlap of transcripts which contain m$^6$A sites in these two cells lines, with 77% of TREx BCBL1-Rta m$^6$A-modified transcripts being present in HepG2 cells. We then investigated the overlap between transcripts which are m$^6$A-modified and bound by SND1 from RIP-seq and eCLIP datasets (*Figure 6b*). This showed an overlap of 1166 transcripts bound by SND1 of which 88% contained m$^6$A sites. We next compared the extent of overlap between m$^6$A and eCLIP peaks on transcripts (*Figure 6c*). We found comparable binding site overlap for SND1 and established readers with m$^6$A peaks, whereas a control protein (TIAL1) showed reduced overlap between m$^6$A peaks and its binding sites. We investigated the repertoire of transcripts bound by SND1 and other reader proteins and found extensive binding to protein coding transcripts (*Figure 6d*). SND1 showed a similar distribution to m$^6$A across transcripts with a bias towards coding region binding in common with most other reader proteins, whereas the control TIAL1 protein displayed a strong 3' UTR bias (*Figure 6e*). We used HOMER to identify motifs bound by SND1 and other established reader proteins using the ENCODE eCLIP datasets. This analysis, using peaks called from all transcripts, revealed a common sequence GGAC, the core of the consensus motif surrounding m$^6$A sites. For SND1, this motif is further enriched from the eight highest scoring motif, when performed on all eCLIP peaks (30,121 peaks), to the second highest scoring motif, upon restricting the analysis to SND1 binding sites on exons which contain the m$^6$A modification (7965 peaks), of which 58% (4,637) directly overlap an m$^6$A site (*Figure 6f*). Together, this analysis reveals that SND1 recognises the consensus m$^6$A motif in vivo and has a similar binding profile to other established reader proteins.

Regarding KSHV viral mRNAs, we were able to confirm *ORF50* RNA as a high-confidence SND1 target. As a comparison, a highly expressed viral lytic RNA, *ORF57*, did not reach the cut-off to be considered a high-confidence target (*Figure 4—figure supplement 3e*). We additionally identified 33, 23 and 14 KSHV transcripts as high-confidence SND1 targets at 0 hr, 8 hr and 20 hr post-reactivation, respectively (*Figure 7a*). Deep-sequencing coverage for high-confidence SND1 targets can be seen in *Figure 7—figure supplements 1* and *2*. Validation of SND1-binding to the different regions containing m$^6$A peaks in the second exon and the intron of *ORF50* RNA was performed by RIP followed by RT-qPCR. A ~ 40 fold SND1 enrichment was detected in the second exon of *ORF50* and a ~ 10 fold enrichment in the intron compared with the SND1 enrichment detected on the non-target *18S rRNA* (*Figure 7b*). We further tested by RIP the binding of endogenous FXR1, FXR2, PSIP1, YTHDF1 and YTHDF3 to *ORF50* RNA, while non-specific rabbit immunoglobulin (IgG) was

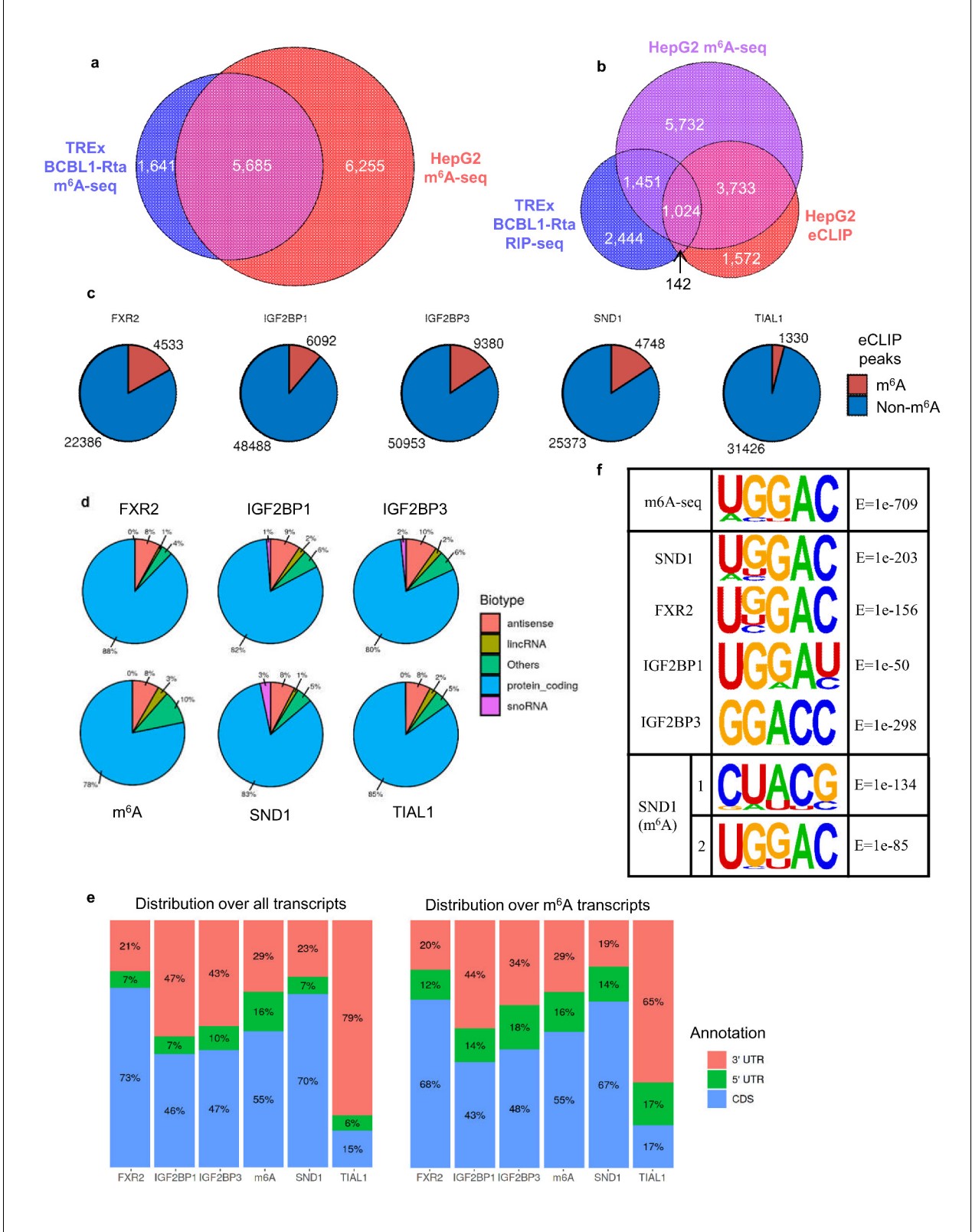

**Figure 6.** SND1 binds the consensus m⁶A motif. (**a**) Overlap of m⁶A-modified transcripts in the indicated cell lines. (**b**) Overlap between the indicated eCLIP, RIP-seq and m⁶A-seq enriched transcripts. (**c**) Direct overlap of m⁶A-seq peaks and eCLIP binding sites for m⁶A reader proteins (FXR2, IGF2BP1, IGF2BP3), SND1 and the control, TIAL1. (**d**) Distribution of binding sites for the indicated proteins across RNA biotypes. (**e**) Distribution of binding sites across different transcript regions for the indicated proteins. (**f**) Motif analysis using HOMER and HepG2 eCLIP datasets as indicated. Top panel m⁶A-

*Figure 6 continued on next page*

*Figure 6 continued*
seq derived motif in HepG2 cells (note that this is the highest scoring motif derived), central panels are consensus motifs found using all target transcripts, bottom panel is the two most highly enriched SND1 motifs in the set of m$^6$A-modified exons bound by SND1.
DOI: https://doi.org/10.7554/eLife.47261.021

used as negative control antibody. The YTHDF1 RNA target *SON* (*Wang et al., 2014*) was used as a binding control RNA for YTH readers. Both YTH readers and FXR2 bound *ORF50* RNA, showing a ~ 8 fold enrichment (*Figure 7b*) and ~15 fold enrichment was observed for FXR1. PSIP1 showed a limited but consistent enrichment (~3 fold) above the negative control IgG. These results highlight that all these Royal members can target *ORF50* RNA, SND1 displaying the highest affinity.

## SND1 stabilises *ORF50* RNA and is essential for KSHV replication

Having established the SND1 RNA-binding topology, we further investigated the role of SND1 in KSHV infection and its relationship with *ORF50* RNA. Here, two TREx BCBL1-Rta cell lines with stable shRNA knockdown of SND1 (SND1 KD1 and SND1 KD2) and a shRNA scramble cell line were generated. Surprisingly, following 24 hr of lytic replication there were no significant differences between the scramble and the knockdown cell lines in the amount of KSHV lytic transcripts produced (*Figure 7c*). In accordance with mRNA levels, RTA and ORF57 proteins were not reduced (*Figure 7d*). Similarly, the early lytic protein ORF54 was not decreased (*Figure 7d*). These data were also confirmed by RNA-seq performed in scramble and SND1 KD2 cells from two biological replicates. Following 24 hr of lytic reactivation, expression of the KSHV transcriptome in SND1-depleted cells was not significantly altered from scramble cells, with the exception of upregulation of *ORF71* and *ORF72* mRNAs (*Figure 7e*). qRT-PCR analysis confirmed a ~ 4 fold-induction of these transcripts in SND1-depleted cells during the lytic cycle (data not shown). These results possibly suggest that SND1 plays a role in maintaining low expression of these latent RNAs during lytic replication. Importantly, in the presence of overexpressed RTA protein, KSHV lytic replication is essentially unchanged in SND1-depleted TREx BCBL1-Rta cells, thus these cells still fulfil all viral functions necessary for triggering the lytic cascade.

TREx BCBL1-Rta cells are reactivated by expression of Myc-tagged RTA which is integrated in the host genome in a spliced form and its expression is under the control of a doxycycline-inducible promoter. In contrast, the parental BCBL1 cell line lacks Myc-tagged RTA and reactivation is achieved by addition of the histone deacetylase inhibitor sodium butyrate (NaB), which leads to the expression of unspliced *ORF50* RNA. This native RNA matures to the spliced form giving rise to RTA protein, which then transactivates the *ORF50* promoter in a positive transcriptional feedback mechanism (*Guito and Lukac, 2012*). However, as SND1 does not take part in the splicing of *ORF50* RNA (*Figure 7c*), it therefore suggests a potential role in the stabilisation of *ORF50* RNA, which could be masked by constitutive overexpression of Myc-tagged RTA in TREx BCBL1-Rta cells. Thus, the same lentiviral shRNAs targeting SND1 were used to knockdown SND1 in BCBL1 cells. Remarkably, both SND1 knockdown cell lines showed a dramatic decrease in viral RNAs, including *ORF50* RNA (*Figure 7f*), and RTA, ORF57 and ORF54 protein levels (*Figure 7g*). These data were also confirmed by RNA-seq performed in scramble and SND1 KD2 BCBL1 cells from two biological replicates. After 24 hr of lytic reactivation, 48 KSHV RNAs were significantly downregulated in SND1-depleted cells compared with scramble cells (FDR < 5%), including *ORF50* RNA (*Figure 7h*). These results indicate that in the absence of SND1 there is a global impairment of lytic KSHV replication downstream of RTA and uncover SND1 as an essential protein for lytic KSHV replication.

Although SND1 did not affect splicing of the *ORF50* RNA, we assessed whether SND1 regulated splicing events in cellular RNA processing, due to the previously reported role of SND1 in the regulation of splicing (*Jariwala et al., 2015*). Thus we hypothesised that SND1 may play a role in splicing through binding methylated intronic regions. Splicing analysis revealed multiple significant differential splicing events between scramble latent cells and scramble lytic TREx BCBL1-Rta cells, however, there were no significant splicing differences between scramble and SND1-depleted cells, when analysing cellular or viral transcripts from either latent or lytic cells (*Figure 8a*). The same results were also observed in BCBL1 cells (data not shown).

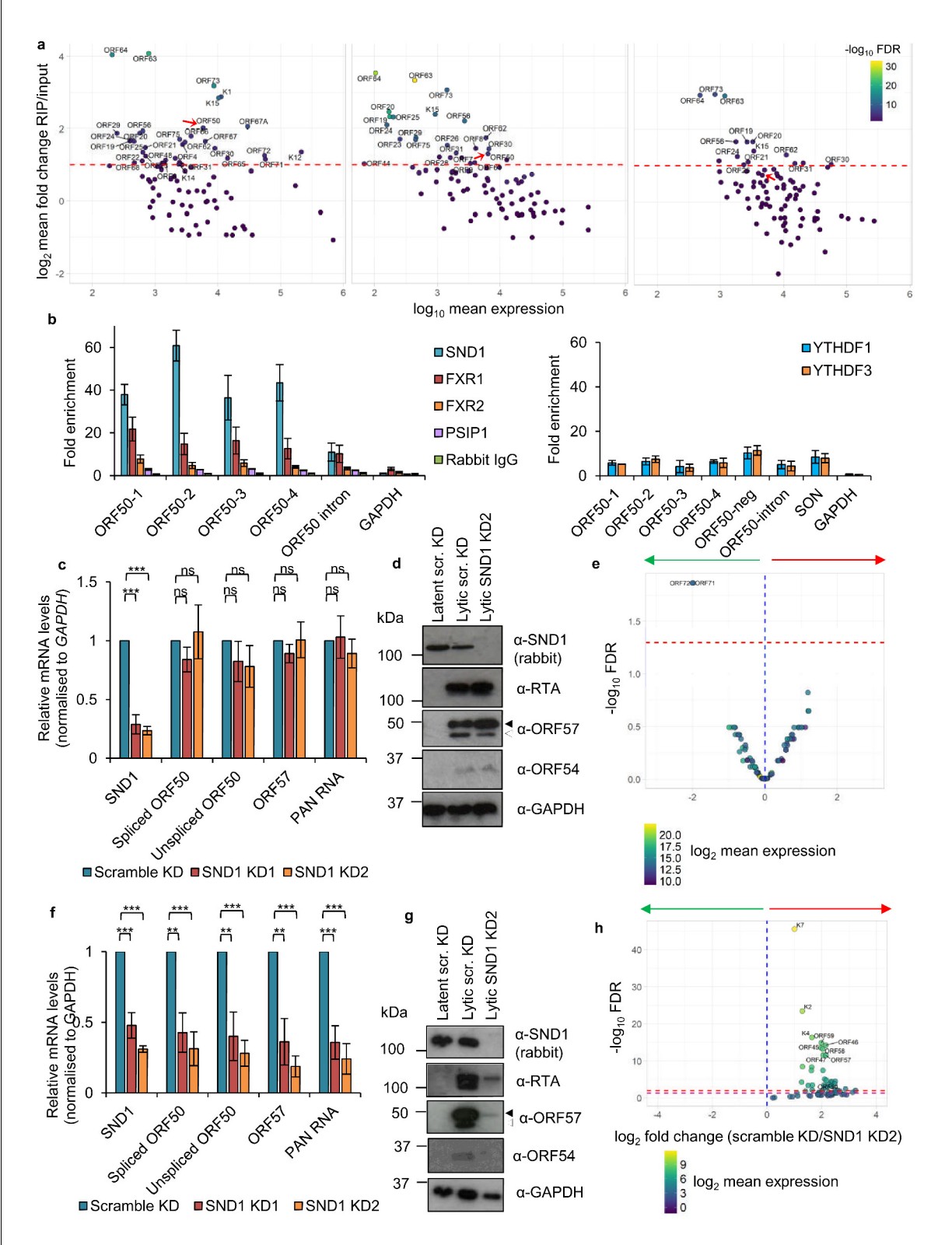

**Figure 7.** SND1 binds the essential *ORF50* RNA in KSHV-infected cells and SND1 knockdown significantly impairs KSHV lytic replication. (a) RIP-seq identifies KSHV mRNAs as high-confidence SND1 targets at 0 hr (left panel), 8 hr (middle panel) and 20 hr (right panel) post-reactivation. Red dashed line indicates a cut-off of >2 fold change RIP/input. Red arrow points *ORF50*. Note that lytic genes are also detected in latent cells (0 hr) due to spontaneous KSHV reactivation and the high sensitivity of deep-sequencing. As expected, significantly lower coverage for these lytic genes was

*Figure 7 continued on next page*

*Figure 7 continued*

detected during latency compared with lytic replication. (**b**) RIP for endogenous SND1, FXR1, FXR2, PSIP1, normal rabbit IgG, YTHDF1 and YTHDF3 followed by qRT-PCR detection. ORF50-1, ORF50-2, ORF50-3 and ORF50-4 indicate primers that generate an amplicon spanning the first, second, third and fourth m6A peaks of the second exon of *ORF50* RNA, respectively. ORF50-intron generates an amplicon spanning the m6A peak in the *ORF50* intron. ORF50 negative primers were designed at the start of the second exon of *ORF50* RNA. Fold enrichment is relative to the enrichment of the non-target *18S rRNA*. Values are averages, error bars present s.d. For SND1 RIPs, n = 4 independent RIPs, for FXR1, FXR2 and normal rabbit IgG n = 3 independent RIPs. For PSIP1 and YTH proteins, n = 2 independent RIPs. (**c–h**) Stable cell lines expressing scramble shRNA (scr. KD) or two independent shRNAs targeting SND1 were generated using TREx BCBL1-Rta cells (**c–e**) or BCBL1 cells (**f–h**). (**c**) Cellular and viral RNA levels were measured by qRT-PCR 24 hr post-reactivation. n = 3 independent viral reactivations. Values are averages, error bars present s.d. ***p<0.001 using an unpaired t-test. ns = not significant. (**d**) Immunoblot analysis of protein lysates from latent or 24 hr reactivated cells. ORF57 antibody detects both full length ORF57 protein (black arrow) and the caspase-7-cleaved form (white arrow). For SND1 detection a rabbit polyclonal antibody was used. Western blots are representative of two independent viral reactivations. (**e**) Volcano plot displaying viral expression of scramble KD and SND1 KD2 cells 24 hr post-reactivation analysed by RNA-seq. The green arrow highlights the quadrant containing upregulated viral ORFs in depleted cells. The red arrow highlights the quadrant containing downregulated ORFs in depleted cells. The red dashed line denotes the FDR < 1% cut-off. (**f**) Cellular and viral RNA levels were measured by qRT-PCR 24 hr post-reactivation. n = 3 independent viral reactivations. Values are averages, error bars present s.d. **p<0.01, ***p<0.001 using an unpaired t-test. (**g**) Immunoblot analysis of protein lysates from latent or 24 hr reactivated cells. Western blots are representative of two independent viral reactivations. (**h**) Volcano plot displaying viral expression of scramble KD and SND1 KD2 cells 24 hr post-reactivation analysed by RNA-seq. The red and purple dashed lines denote the FDR < 1% and FDR < 5% cut-offs, respectively.
DOI: https://doi.org/10.7554/eLife.47261.022

The following source data and figure supplements are available for figure 7:

**Source data 1.** Source data for qRT-PCR experiments and uncropped western blots.
DOI: https://doi.org/10.7554/eLife.47261.028
**Figure supplement 1.** Sequencing coverage tracks for SND1 high-confidence viral RNA targets at 20 hr post-reactivation.
DOI: https://doi.org/10.7554/eLife.47261.023
**Figure supplement 2.** Sequencing coverage tracks for SND1 high-confidence viral RNA targets at 20 hr post-reactivation.
DOI: https://doi.org/10.7554/eLife.47261.024
**Figure supplement 3.** SND1 does not bind the KSHV *ORF50* promoter.
DOI: https://doi.org/10.7554/eLife.47261.025
**Figure supplement 4.** m6A methylation has a pro-viral role in the KSHV lytic life cycle.
DOI: https://doi.org/10.7554/eLife.47261.026
**Figure supplement 5.** Depletion of YTH readers impairs KSHV lytic replication.
DOI: https://doi.org/10.7554/eLife.47261.027

As SND1 has previously been described as a transcriptional activator (*Jariwala et al., 2015*) we determined whether it could associate with the *ORF50* promoter by carrying out chromatin immuno-precipitation (ChIP) with the ChIP-grade antibody we previously used for RIP-seq experiments. RNA polymerase II (RNAPII) antibody (clone CTD4H8) and non-specific immunoglobulin (IgG) were used respectively as positive and negative control antibodies. Whilst RNAPII was enriched at the *GAPDH* promoter and multiple viral promoters including *ORF50*, there was no significant enrichment for either SND1 or the non-specific IgG (*Figure 7—figure supplement 3*). Moreover, RNA-seq analysis of TREX BCBL1-Rta-SND1 depleted cells showed no significantly downregulated viral gene expression (*Figure 7e*). Consequently, we conclude that SND1 does not participate in the activation of the *ORF50* promoter.

The strikingly different knockdown phenotype between TREX BCBL1-Rta and BCBL1 cells directly points to a potential regulatory role of SND1 on the essential lytic *ORF50* RNA, and suggests that SND1 may stabilise the *ORF50* RNA. To test this hypothesis without the possible interference of NaB on *ORF50* RNA decay, the decay of native (unspliced) *ORF50* RNA was monitored in scramble and SND1-depleted TREx BCBL1-Rta cells that had been reactivated into the lytic cycle for 24 hr before addition of actinomycin D. Strikingly, native *ORF50* RNA was significantly more unstable in SND1-depleted cells (*Figure 8b*) with ~80% of *ORF50* RNA remaining at 6 hr post-transcription inhibition in scramble cells and ~50% in depleted cells. To further determine whether SND1 may also play a role in regulating the stability of *ORF71* and *ORF72* RNAs during the KSHV lytic cycle, the turnover of these transcripts together with other high confidence SND1 KSHV RNA targets was also investigated. In contrast to *ORF50* RNA, these transcripts decayed in a similar manner in scramble and depleted cells (*Figure 8b*).

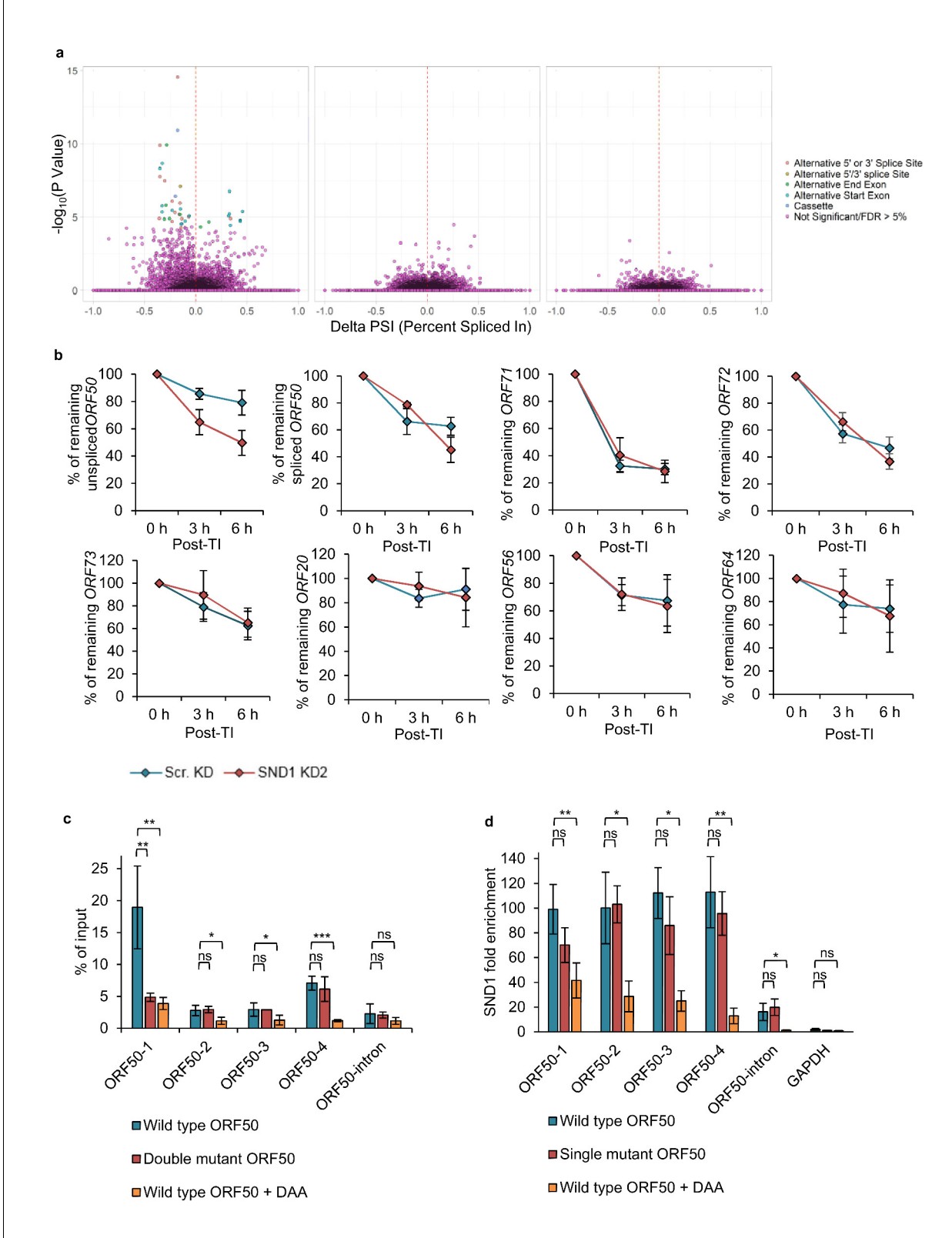

**Figure 8.** SND1 stabilises native *ORF50* RNA and inhibition of m[6]A deposition on this RNA abolishes SND1 binding. (a) RNA-seq analysis reveals significant alterations in splicing events between scramble latent and scramble lytic TREx BCBL1-Rta cells (left panel), whilst no significant changes in splicing are observed between scramble and SND1-depleted cells during latency (middle panel) or 24 hr lytic replication (right panel). (b) TREx BCBL1-Rta cells were reactivated for 24 hr into the lytic cycle and transcription was inhibited with the addition of actinomycin D (2.5 μg/ml). Transcripts of

*Figure 8 continued on next page*

*Figure 8 continued*

interest were measured by qRT-PCR at 0 hr, 3 hr and 6 hr post-transcription inhibition (TI). Each viral gene was normalised against *18S rRNA*. Values are averages, error bars present s.d. n = 4 independent viral reactivations. (c) m$^6$A-enrichment was determined by m$^6$A-IP-qPCR. HEK 293 cells were transfected for 24 hr with either wild type (WT) FLAG-ORF50 or a double mutant plasmid in which the GGACT motifs present in the ORF50-1 and ORF50-4 baits were mutated to GGATT. When using DAA, 4 hr after transfection DAA was added at a concentration of 200 µM, and 24 hr post-treatment cells were harvested. % of input was calculated similarly as to ChIP-qPCR analysis. Values are averages, error bars present s.d. *p<0.05, **p<0.01, ***p<0.001 using an unpaired t-test. ns = not significant. For WT ORF50 n = 9 independent m$^6$A-IPs [3 including 0.25% (v/v) DMSO-treatment), for double mutant ORF50 n = 3, for DAA-treated cells n = 3. (d) HEK 293 cells were transfected for 24 hr with either wild type (WT) FLAG-ORF50 or a single mutant plasmid in which the GGACT motif present in the ORF50-1 bait was mutated to GGATT. For DAA-treated cells, 4 hr after transfection DAA was added at a concentration of 200 µM, and 24 hr post-treatment cells were harvested. SND1 enrichment was determined by SND1-RIP-qPCR and is relative to the enrichment found in the non-target *18S rRNA*. *GAPDH* RNA served as an additional non-target RNA. Values are averages, error bars present s.d. For WT and mutant ORF50 n = 4 independent RIPs. For DAA-treated cells n = 3 independent RIPs. *p<0.05, **p<0.01 using an unpaired t-test.

DOI: https://doi.org/10.7554/eLife.47261.029

The following source data and figure supplement are available for figure 8:

**Source data 1.** Source data for qRT-PCR experiments.

DOI: https://doi.org/10.7554/eLife.47261.031

**Figure supplement 1.** DAA was not cytotoxic in HEK-293 cells.

DOI: https://doi.org/10.7554/eLife.47261.030

## Removal of m$^6$A in *ORF50* RNA impairs SND1 binding to *ORF50* RNA

Due to the broad and overlapping m$^6$A-seq peaks across the *ORF50* RNA and the high frequency of DRm$^6$ACH motifs throughout *ORF50* RNA, we attempted to deplete global m$^6$A levels in *ORF50* RNA by stably depleting METTL3 in BCBL1 cells and assess its effect on lytic replication, however limited depletion was achieved and no significant effect on KSHV lytic replication was observed (data not shown). The same METTL3 knockdowns were repeated in BCBL1 cells but transiently. After five days post-lentiviral transduction, cells were reactivated for 24 hr and viral and protein levels analysed. Here, one METTL3 knockdown cell line (METTL3 KD2) achieved a higher level of depletion of METTL3 protein and viral transcript and protein levels were both significantly reduced compared with the scramble cell line (*Figure 7—figure supplement 4a and b*). In contrast, a second cell line (METTL3 KD1) showed no depletion of METTL3 protein and no significant differences in viral replication between these cells and the scramble were observed (*Figure 7—figure supplement 4a and b*).

In addition, we generated stable BCBL1 cell lines harbouring shRNA knockdown of either FTO, YTHDF1 or YTHFD3. In contrast, despite consistently achieving ~75% depletion of YTHDF2 at the mRNA level, following 12 days post-transduction these depleted cells would dramatically lose their knockdown at the mRNA level, suggesting that YTHDF2 may be essential for BCBL1 cells, therefore we performed transient transductions for YTHDF2. Following reactivation, FTO-depleted cells displayed increased viral mRNA and protein levels (*Figure 7—figure supplement 4c and d*), including increased levels of *ORF50* RNA and RTA protein. Depletion of YTHDF1 or YTHDF3 readers resulted in decreased *ORF50* RNA together with other lytic RNAs (*Figure 7—figure supplement 5a-b and e-f*), particularly, in YTHDF1-depleted cells, which also showed a marked reduction of viral protein levels. Depletion of YTHDF2 did not affect viral RNA levels, with the exception of PAN RNA, however a clear decrease in RTA protein and a slight reduction in ORF57 protein levels were evident (*Figure 7—figure supplement 5c and d*). Taken together, these results indicate that m$^6$A modification has a pro-viral role in the KSHV lytic replication cycle, in agreement with previous studies performed in the same cell line (*Ye et al., 2017*).

Next, we set out to determine whether the m$^6$A status of *ORF50* RNA regulates SND1 binding. Firstly, single-point mutations were performed in the GGACU motifs present in ORF50 baits to elucidate whether the chosen motifs were m$^6$A-modified in the context of the full length *ORF50* RNA. Wild type (WT) FLAG-ORF50 -containing both ORF50 exons and the intron- and mutant plasmids were transfected into HEK-293 cells and m$^6$A enrichment quantified by m$^6$A-IP-qPCR (*Figure 8c*). In transfected cells, *ORF50* RNA was particularly m$^6$A-modified in ORF50-1 and ORF50-4 regions, suggesting cell type differences between TREx BCBL1-Rta and HEK-293 cells. Point mutation in the motif contained in ORF50-1 bait, but not in ORF50-4 bait, significantly reduced m$^6$A enrichment, confirming that this site is m$^6$A-modified. Next, the binding of SND1 to WT FLAG-ORF50 and to a

FLAG-ORF50 plasmid with a point mutation in the motif contained in ORF50-1 bait was evaluated by RIP. No significant decrease in SND1 binding was observed (*Figure 8d*), indicating that other SND1 binding sites are necessary for *ORF50* RNA-SND1 interaction.

We finally assessed the binding of SND1 to *ORF50* RNA in the absence or presence of $m^6A$ modification. For this purpose, we made use of the drug 3-deazaadenosine (DAA), which inhibits $m^6A$ deposition by reducing levels of the methyl donor S-adenosylmethionine (SAM). HEK-293 cells were transfected with WT FLAG-ORF50 plasmid and 4 hr after transfection, control 0.25% (v/v) DMSO or 200 µM DAA was incubated for an additional 24 hr. Note that DAA did not result in cytotoxicity at this concentration after 26 hr post-treatment (*Figure 8—figure supplement 1*). DAA effectively reduced $m^6A$ enrichment in all $m^6A$-modified regions of *ORF50* RNA (*Figure 8c*) and led to a significant decrease of SND1 binding across *ORF50* RNA (*Figure 8d*). Interestingly, DAA completely abolished SND1 binding to the native *ORF50* transcript, indicating that $m^6A$ inhibition on *ORF50* RNA regulates SND1 binding to it, however the exact points of interaction between SND1 and *ORF50* RNA remain to be elucidated at single nucleotide resolution.

In summary, these experiments propose a model where in the absence of SND1, unspliced *ORF50* RNA is more unstable resulting in reduced RTA protein levels which will further reduce activation of the RTA promoter in a negative feedback loop that culminates in lytic replication impairment.

## Discussion

SND1 is a multi-functional protein. It functions as a transcriptional co-activator of Epstein-Barr virus nuclear protein 2, STAT5, STAT6 and c-Myb (*Jariwala et al., 2015*). Additionally, SND1 has multiple roles modulating gene expression at a post-transcriptional level. SND1 is a component of the RISC complex (*Jariwala et al., 2015*) and acts in the processing of specific miRNAs (*Heinrich et al., 2013*). $m^6A$ promotes processing of primary miRNAs (pri-miRNAs) (*Alarcón et al., 2015*), thus SND1 may also process $m^6A$-modified pri-miRNAs, which could explain the SND1-binding in methylated introns we observed. In agreement with our finding that SND1 stabilises *ORF50* RNA, SND1 binds to the 3′UTR of angiotensin II type one receptor (AT1R) and stabilises this mRNA leading to elevated protein AT1R levels (*Jariwala et al., 2015*). Intriguingly, *AT1R* mRNA contains a single $m^6A$ site which is located in the 3′UTR region (*Sun et al., 2016*). Finally, SND1 also participates in RNA editing (*Jariwala et al., 2015*). Under stress conditions, in mammalian cells SND1 co-localises with G3BP protein in stress granules (*Gao et al., 2010*) and stabilises *AT1R* and *IGFBP2* mRNAs (*Gao et al., 2015*). Similarly, in plants SND1 is essential for the stabilisation of a subset of stress-responsive mRNAs (*Frei dit Frey et al., 2010*). It will be of interest to address to what extent SND1 regulates the stability of its target RNAs. Curiously, the readers YTHDF1-3, FXR1, FXR2 and FMR1 have also been identified in mammalian stress granule cores (*Jain et al., 2016*).

We performed RNA-lifetime profiling in scramble and SND1-depleted cells using latent and lytic TREx BCBL1-Rta and BCBL1 cells; however the NGS data were extremely noisy and this precluded us from identifying a consistent phenotype between replicates.

To date, very little research has been performed on deciphering the role of SND1 during viral infections. SND1 binds the 3′UTRs of transmissible gastroenteritis coronavirus (TGEV) (*Galán et al., 2009*) and dengue virus (DENV). Moreover, SND1 silencing reduced the levels of viral RNA and protein in DENV-infected cells (*Lei et al., 2011*). Intriguingly, the 3′UTR of DENV contains $m^6A$ sites (*Gokhale et al., 2016*), thus it would be of considerable interest to examine whether this modification enables SND1 recruitment to the 3′UTR to stabilise the RNA DENV genome.

Differences in the binding affinities for the different RNA baits used in this study between YTH readers and the Royal members can be inferred from previous structural studies. Interestingly, in SND1 (*Chen et al., 2009a*), plant agenet members (*Adams-Cioaba et al., 2010*), PSIP1, HDGFRP2 and MSH6 (*Qin and Min, 2014*), the aromatic pocket is hydrophobic and surrounded by both positive and negative residues thus, it seems plausible that binding only ensues when matching electrostatic interactions are achieved between the RNA sequence and the Royal domain, consequently, methylated proteins would adopt a different orientation to interact with negatively charged residues while $m^6A$-decorated RNAs would interact with positively charged residues. This model would explain the RNA structure dependency observed for these proteins in binding $m^6A$-modified RNA. In contrast, the aromatic cage of YTH readers resides in a hydrophobic pocket surrounded by a

positively charged surface, rendering the YTH domain favourable to bind any methylated RNA sequence. Of note, plant agenet members contain two plant agenet domains (Agenet1 and Agenet2) in tandem, each harbouring an aromatic cage. Whilst the aromatic cage of Agenet2 is proposed to be the site that binds methylated lysines (*Adams-Cioaba et al., 2010*), Agenet1 exhibits a basic surface patch with potential for RNA binding (*Myrick et al., 2015*). The plant Agenet domain of FMRP (1-134) has already been shown to harbour RNA-binding ability to RNA homopolymers, with progressively increased affinity when testing longer FMRP constructs (1–180 and 1–214), suggesting a cooperative effect between the positively charged residues distributed along the N-terminus region (*Milosevich and Hof, 2016*). In a similar manner, the other domains present in SND1 cannot be disregarded in considering how SND1 may bind its target RNAs. Our RIP-seq data and eCLIP analysis reveals for the first time that SND1 is indeed a *bona fide* RNA-binding protein acting at the transcriptome level. Structural and biochemical analysis had already proposed that the N-terminal region of SND1, specifically SN3 and SN4, possess RNA binding (*Li et al., 2008*), thus, in addition to the Tudor domain interacting with methylated RNA, the N-terminal region of SND1 most likely also contributes to RNA binding and may play a significant role in determining which RNAs are targeted by SND1, including those that are methylated.

Elucidating Royal domains structures in complex with $m^6$A-ORF50-1 hairpin will help reveal the reason for the distinct selectivity between YTH readers and Royal members and to elucidate to which extent the aromatic cage of Royal domains plays a role in recognising $m^6$A-modified RNA. Moreover, these studies could lead to the development of small molecule inhibitors that specifically block the aromatic cage of SND1 to hinder recognition of methylated *ORF50* RNA for the treatment of KSHV-related malignancies. Pioneering inhibitors for some methyl-lysine readers of the Tudor subfamily and other Royal members are currently being investigated with success (*Milosevich and Hof, 2016*).

Our findings underscore the potential of other members from the 'Royal family' as putative new regulators of $m^6$A. Of the Tudor subfamily, the Tudor domain-containing (TDRD) proteins, AKAP1, SMN and SPF30 display methyl-arginine-binding and are involved in RNA metabolism (*Chen et al., 2011*), therefore these are ideal candidates to reveal more $m^6$A readers. In contrast to the ubiquitously expressed SND1, most of the mammalian TDRD proteins display male germline-enriched expression and are essential for spermatogenesis (*Chen et al., 2011*). It will be of interest to examine whether $m^6$A-decorated RNAs are regulated post-transcriptionally via TDRD proteins during germ cell differentiation. Notably, spermatid perinuclear RNA-binding protein (STRBP) was within the top ten enriched proteins in $m^6$A-ORF50-1 bait. The remainder members of the Tudor protein subfamily bind methyl-lysine residues and contain other domains related to chromatin biology, consequently, these proteins are unlikely to participate in RNA metabolism. The discovery of PSIP1, HDGFRP2 and MSH6 as putative $m^6$A readers is surprising as the PWWP domain is a well-established nucleosome-binding domain and these proteins participate in DNA repair (*Qin and Min, 2014*). However, FMRP takes part in the DNA damage response (*Alpatov et al., 2014*) but FMRP is also a RNA-binding protein that regulates the translation of its $m^6$A-containing target RNAs (*Edupuganti et al., 2017*). An $m^6$A RNA-mediated response to UV-induced DNA damage was recently reported (*Xiang et al., 2017*), as such, these PWWP proteins could represent a link between methylation of RNA and DNA damage. In support of the putative $m^6$A reading ability of these proteins is the fact that mass spectrometry identification of PSIP1 short (p52) isoform, which includes the PWWP domain, revealed that ~95% of interactors function in pre-mRNA processing. Moreover this isoform co-localised with SRSF2 in nuclear speckles and modulated alternative splicing (*Pradeepa et al., 2012*), thus the implication of these proteins in $m^6$A RNA metabolism requires further investigation.

Finally, it is worth pointing out the possibility that SND1 may be able to read other RNA methyl-modifications in addition to $m^6$A. Further studies characterising these methyl-modifications and their corresponding readers will help resolve this matter.

In conclusion, our data supports the hypothesis that highly specialised domains such as the Royal domains, which harbour a structurally-related aromatic cage to the one found in the YTH domain, may be required for the selective and direct recognition of $m^6$A, while proteins without aromatic cages may not be able to directly read $m^6$A.

# Materials and methods

## Key resources table

| Reagent type (species) or resource | Designation | Source or reference | Identifiers | Additional information |
|---|---|---|---|---|
| Strain, strain background (*Escherichia coli*) | BL21(DE3) | Thermo Scientific | Cat No. C600003 | Competent cells |
| Cell line (*Homo sapiens*) | HEK-293T | ATCC | Cat No. CRL-3216 | The cell line is commercially available at ATCC |
| Cell line (*Homo sapiens*) | HEK-293 | ATCC | Cat No. CRL-1573 | The cell line is commercially available at ATCC |
| Cell line (*Homo sapiens*) | TREx BCBL1-Rta | A gift of JU Jung (University of Southern California, USA). | *Nakamura et al., 2003* | |
| Cell line (*Homo sapiens*) | BCBL1 | A gift from Dr Andrew Hislop (University of Birmingham, UK). | *Renne et al., 1996* | |
| Antibody | Anti-m$^6$A (rabbit polyclonal) | Merck Millipore | ABE572 | m$^6$A-seq and m$^6$A-IP |
| Antibody | Anti-SND1 (rabbit polyclonal) | Proteintech | 10760–1-AP | WB (1:1,000). 2 µg for RIP-seq and RIP 2 µg per ChIP |
| Antibody | Anti-SND1 (mouse monoclonal) | Proteintech | 60265–1-Ig | WB (1:1,000). |
| Antibody | Anti-FXR1 (rabbit polyclonal) | Proteintech | 13194–1-AP | 2 µg per RIP |
| Antibody | Anti-FXR2 (rabbit polyclonal) | Proteintech | 12552–1-AP | 2 µg per RIP |
| Antibody | Anti-PSIP1 (rabbit polyclonal) | Proteintech | 25504–1-AP | 2 µg per RIP |
| Antibody | Anti-METTL3 (rabbit polyclonal) | Bethyl laboratories | A301-567A | WB (1:500) |
| Antibody | Anti-FTO (rabbit monoclonal) | Abcam | ab126605 | WB (1:5,000) |
| Antibody | Anti-YTHDF1 (rabbit polyclonal) | Proteintech | 17479–1-AP | WB (1:1,000) 1 µg per RIP |
| Antibody | Anti-YTHDF2 (rabbit polyclonal) | Abclonal | A9639 | WB (1:500) |
| Antibody | Anti-YTHDF3 (rabbit polyclonal) | Abclonal | A8395 | WB (1:500) 5.8 µg per RIP |
| Antibody | Anti-ORF57 (mouse monoclonal) | Santa Cruz | sc-135746 | WB (1:1,000) |
| Antibody | Anti-RTA (rabbit polyclonal) | A gift from Professor David Lukac (Rutgers, New Jersey, USA) | *Lukac et al., 1998* | WB (1:1,000) |

*Continued on next page*

*Continued*

| Reagent type (species) or resource | Designation | Source or reference | Identifiers | Additional information |
|---|---|---|---|---|
| Antibody | Anti-ORF54 (mouse monoclonal) | A gift from Friedrich Grässer (University of Homburg, Germany) | *Kremmer et al., 1999* | WB (1:1,000) |
| Recombinant DNA reagent | FLAG-*ORF50* plasmid (pCDH-CMV-MCS-EF1-Puro) for mammalian expression | NovoPro Bioscience | Custom made. | Purchased from NovoPro, available upon request from the Whitehouse laboratory. |
| Recombinant DNA reagent | GST-SND1-C-terminus (residues 548–910) for bacteria expression | NovoPro Bioscience | Custom made. | Purchased from NovoPro, available upon request from the Whitehouse laboratory. |
| Recombinant DNA reagent | GST-FXR1-plant agenet (residues 2–132) for bacteria expression | NovoPro Bioscience | Custom made. | Purchased from NovoPro, available upon request from the Whitehouse laboratory. |
| Recombinant DNA reagent | GST-PSIP1-PWWP (residues 3–100) for bacteria expression | NovoPro Bioscience | Custom made. | Purchased from NovoPro, available upon request from the Whitehouse laboratory. |
| Recombinant DNA reagent | GST-CBX3-Chromo (residues 29–86) for bacteria expression | NovoPro Bioscience | Custom made. | Purchased from NovoPro, available upon request from the Whitehouse laboratory. |
| Sequence-based reagent | Mission TRC shRNA SND1 KD1 | Sigma | TRCN0000245143 | Mission TRC shRNA bacterial glycerol stock commercially available from Sigma |
| Sequence-based reagent | Mission TRC shRNA SND1 KD2 | Sigma | TRCN0000049656 | Mission TRC shRNA bacterial glycerol stock commercially available from Sigma |
| Commercial assay or kit | RNA fragmentation reagent | Thermo Scientific | AM8740 | |
| Commercial assay or kit | Pierce chromatin prep module | Thermo Scientific | 26158 | |
| Commercial assay or kit | LightShift chemiluminescent RNA EMSA Kit | Thermo Scientific | 20158 | |
| Commercial assay or kit | EZ-ChIP | Merck Millipore | 17–371 | |
| Commercial assay or kit | DNA-*free* DNA Removal Kit | Thermo Scientific | AM1906 | |
| Chemical compound, drug | 3-deazaadenosine (DAA) | Cambridge Bioscience | 9000785–5 mg | |
| Software, algorithm | m6aViewer 1.6 software | dna2.leeds.ac.uk/m6a/ | *Antanaviciute et al., 2017* | Published in the RNA journal. 2017 Oct; 23(10): 1493–1501. |
| Other | Magna ChIP Protein A+G magnetic beads | Merck Millipore | 16–663 | Used for m$^6$A-seq and RIP-seq |

## Cell lines and reagents

HEK-293T and HEK-293 cells were purchased from ATCC (American Type Culture Collection) and cultured in Dulbecco's modified Eagle's medium with glutamine (DMEM, Lonza) supplemented with 10% (v/v) fetal calf serum (FCS) (Gibco) and 1% (v/v) penicillin-streptomycin (P/S) (Gibco). TREx BCBL1-Rta cells, a BCBL1-based, primary effusion lymphoma (PEL) B cell line that has been engineered to inducibly express exogenous Myc-tagged RTA by the addition of doxycycline, were a gift of JU Jung (University of Southern California, USA). BCBL1 cells were a gift from Dr Andrew Hislop (University of Birmingham, UK). BCBL1 cells were grown in RPMI1640 growth medium with glutamine (Gibco) supplemented with 10% (v/v) FCS (Gibco) and 1% (v/v) P/S (Gibco). TREx BCBL1-Rta cells were grown in RPMI1640 growth medium with glutamine (Gibco) supplemented with 10% (v/v) FCS, (Gibco), 1% P/S (v/v) (Gibco) and 100 µg/mL hygromycin B (Thermo Scientific). All cell lines were tested negative for mycoplasma. For virus reactivation, TREx BCBL1-Rta cells were induced using 2 µg/mL doxycycline hyclate (Sigma-Aldrich) and BCBL1 cells were induced using 2 mM sodium butyrate (Sigma-Aldrich). Antibodies used in Western blotting are listed below: anti-SND1 (Proteintech, 10760–1-AP, 1:1,000); anti-SND1 (Proteintech, 60265–1-Ig, 1:1,000); anti-METTL3 (Bethyl, A301-567A, 1:500); anti-FTO (Abcam, ab126605, 1:5,000); anti-YTHDF1 (Proteintech, 17479–1-AP, 1:1,000); anti-YTHDF2 (Abclonal A9639, 1:500); anti-YTHDF3 (Abclonal, A8395, 1:500); anti-ORF57 (Santa Cruz, sc-135746 1:1,000); anti-GAPDH (Abcam, ab8245 1:5,000); anti-PARP (CST, 9542 1:2,500), the rabbit polyclonal anti-RTA was a gift from Professor David Blackbourn (University of Surrey, UK) and used at 1:1000. The mouse monoclonal anti-ORF54 was a gift from Friedrich Grässer (University of Homburg, Germany) and used at 1:1000. Rabbit anti-m$^6$A antibody (ABE572) (Merck Millipore) was used in m$^6$A-immunoprecipitations. Antibodies used in RIP are as follows: anti-SND1 (Proteintech, 10760–1-AP); anti-FXR1 (Proteintech, 13194–1-AP); anti-FXR2 (Proteintech, 12552–1-AP); anti-PSIP1 (Proteintech, 25504–1-AP); anti-YTHDF1 (Proteintech, 17479–1-AP); anti-YTHDF3 (Abclonal, A8395) and normal rabbit IgG (Merck Millipore, 12–370). 3-deazaadenosine (DAA) was purchased from Cambridge Bioscience. For RIPs, either 2 µg of anti-SND1, FXR1, FXR2, PSIP1 or normal rabbit IgG were used per RNA immunoprecipitation. For YTHDF1 and YTHDF3, 1 and 5.8 µg were used per immunoprecipitation respectively. For ChIP experiments, 2 µg of α-RNAPII (clone CTD4H8) antibody (Merck Millipore), anti-SND1 (Proteintech, 10760–1-AP) or normal rabbit IgG (Merck Millipore, 12–370) were used per chromatin immunoprecipitation. The same lot number (00020506) of SND1 antibody (Proteintech, 10760–1-AP) was used for western blot, RIP-seq, RIP-qPCR and ChIP experiments.

## m$^6$A-seq

Total RNA from TREx BCBL1-Rta cells was extracted using TRIzol (Thermo Scientific) according to the supplier's protocol. DNA-*free* DNA Removal Kit (Ambion) was used to remove any contaminating DNA from RNA samples. Isolated RNA was purified by standard ethanol precipitation and 100 µg of total RNA was fragmented with RNA fragmentation reagent (Ambion) according to the manufacturer's protocol. Fragmented RNA was ethanol-precipitated and re-suspended in 10 µl of RNase-free water and stored at −80°C. 2 µg of total fragmented RNA was saved as input RNA for later use in cDNA library construction. For each m$^6$A-immunoprecipitation (m$^6$A-IP), 25 µl of slurry of Magna ChIP Protein A+G magnetic beads (Merck Millipore) were washed twice with IP/wash buffer [20 mM Tris HCl pH 7.4, 150 mM NaCl, and 0.1% NP-40 (v/v)]. Beads were re-suspended in 100 µl of IP/wash buffer and coated with 5 µg of rabbit anti-m$^6$A antibody (ABE572) (Merck Millipore) for 45 min at room temperature with rotation. Beads were then washed three times with IP/wash buffer and IPs were prepared by mixing the antibody-coated beads with 900 µl of IP/wash buffer, 35 µl of 0.5 M EDTA pH 8.0, 4 µl of *RNasin Plus* (Promega) and 100 µg of fragmented RNA. IPs were incubated overnight at 4°C with rotation. Beads were then washed six times with IP/wash buffer. IP samples were further incubated with 126 µl of IP/wash buffer, 15 µl of 10% SDS (v/v) and 9 µl of PCR-grade proteinase K (20 mg/mL) (Thermo Scientific) for 30 min at 55°C. After incubation, the supernatant containing the RNA was transferred to a new microcentrifuge tube and 250 µl of IP/wash buffer was added to each sample. RNA was purified with the use of phenol:chloroform:isoamyl alcohol (Sigma-Aldrich) and finally sodium acetate/ethanol-precipitated together with 1 µl of RNA-grade glycogen (Thermo Scientific) to allow visualisation of the RNA pellet. Several m$^6$A-IPs (four to six) were pooled to provide enough sample for cDNA library construction and next-generation sequencing (NGS) as

described below. 1.5 to 3 ng of RNA from input and m[6]A-IPs were used for NGS library production using NEBNext Ultra kit (NEB) according to the manufacturer's protocol. Libraries for the first biological replicate were sequenced on a HiSeq 2500 platform (Illumina) with 101 bp paired-end lane. Libraries from the second biological replicate were sequenced on a HiSeq 3000 platform (Illumina) with 151 bp paired-end lane. For m[6]A-seq two independent biological replicates were prepared for each time point analysed (0 hr, 8 hr and 20 hr post-reactivation).

## RIP-seq

TREx BCBL1-Rta cells remained unreactivated or were reactivated for 8 or 20 hr. At the desired time point a fraction of the cells was removed to serve as input RNA to control for RNA expression and stored in TRIzol (Thermo Scientific) at −80°C. For each RNA immunoprecipitation (RIP), $7 \times 10^6$ cells were used. Cells were fixed with 1% (v/v) formaldehyde (Calbiochem) for 10 min at room temperature. Crosslinking was stopped by addition of glycine at a final concentration of 125 mM for 5 min. Cells were washed with PBS (Lonza) and re-suspended in 200 µl of ice-cold shearing buffer [50 mM Tris HCl pH 7.4, 100 mM NaCl and 0.1% (v/v) NP-40] supplemented with Complete, EDTA-free protease inhibitors (Roche) and 1 µl of murine RNase inhibitor (NEB). Samples were then sonicated with an EpiShear multi-sample sonicator (Active Motif) with pulses of 30 s of sonication followed by a 30 s rest for a total of 12 min at 30% amplitude. Polystyrene sonication tubes (Active Motif) were used to achieve efficient sonication. After sonication, samples were centrifuged at 12,000 x $g$ for 10 min at 4°C and the supernatant was used immediately in RIPs. For each RIP, 25 µl of slurry of Magna ChIP Protein A+G magnetic beads (Merck Millipore) were coated with the antibody of interest as previously described for m[6]A-seq. RIPs were prepared by mixing the antibody-coated beads with 800 µl of IP/wash buffer [20 mM Tris HCl pH 7.4, 150 mM NaCl, and 0.1% NP-40 (v/v)], 35 µl of 0.5 M EDTA pH 8.0, 4 µl of murine RNase inhibitor (NEB) and 200 µl of lysate containing fragmented RNA. RIPs were incubated for 3 hr at 4°C with rotation. Beads were then washed six times as previously described (*Gilbert and Sj, 2006*). RIP samples were then further incubated with 200 µl of IP/wash buffer, 2 µl of PCR-grade proteinase K (20 mg/mL) (Thermo Scientific), 4 µl of 5M NaCl and 0.5 µl of murine RNase inhibitor (NEB) for 1 hr at 60°C. After incubation, the supernatant containing the RNA was transferred to a new microcentrifuge tube and 50 µl of IP/wash buffer was added to each sample. RNA was purified as described for m[6]A-seq but instead of using phenol:chloroform:isoamyl alcohol for purification, TRIzol LS (Thermo Scientific) was used according to the manufacturer's instructions. DNA-*free* DNA Removal Kit (Ambion) was used to remove any contaminating DNA from RIP RNA samples. Total RNA from input samples was isolated with the use of TRIzol (Thermo Scientific) according to the supplier's protocol and treated with DNase I using the DNA-*free* DNA Removal Kit (Ambion). Input RNA was saved without applying sonication because we observed lower quality sequencing libraries when using sonicated input RNA than when using input RNA that was heat-fragmented. Saved input RNA was therefore heat-fragmented for 8 min at 94°C before proceeding with library construction. Due to the large size of RNA fragments observed after SND1 immunoprecipitation, isolated RNA from RIP samples was further fragmented for 7 min at 94°C before first strand synthesis. 100 to 200 ng of each input and RIP sample was used for NGS libraries which were made using the TruSeq Stranded Total RNA library production kit (Illumina) according to the manufacturer's protocol. All samples were multiplexed and sequenced on two 151 bp paired-end lanes on a HiSeq 3000 instrument (Illumina). For RIP-seq, two independent biological replicates were prepared for each time point analysed (0 hr, 8 hr and 20 hr post-reactivation). Additionally, for one biological replicate, two technical replicates were deep-sequenced for all RIP samples. A third biological replicate RIP sample at 0 hr was also deep-sequenced.

## RNA-seq

NGS libraries were generated from two independent biological replicates using scramble and SND1 KD2 TREx BCBL1-Rta cells both during latency and after 24 hr of lytic reactivation. Two independent biological replicates from scramble and SND1 KD2 BCBL1 cells both during latency and after 24 hr of lytic reactivation were also deep-sequenced. Libraries were made using TruSeq Stranded Total RNA library preparation kit (Illumina) according to the manufacturer's protocol. Libraries were sequenced on 151 bp paired-end lanes on a HiSeq 3000 instrument (Illumina).

## m⁶A-IP-qPCR and RIP-qPCR

m⁶A-IPs and RIPs were carried out as described above respectively, with the following modification. 1% input samples (10 μl from the 1 mL IP reaction) were removed before immunoprecipitation and stored at −80°C. Input samples were processed together with immunoprecipitated samples from the proteinase K treatment onwards. Purified input and immunoprecipitated RNA was resuspended in 10 μl of RNase-free water and reverse transcribed as described in the RT-qPCR analysis section. qPCR normalisation was performed similarly to chromatin immunoprecipitation (ChIP) coupled to detection by qPCR analysis using ΔΔCt method (relative quantification). An example for m⁶A-IPs follows. Each m⁶A-IP sample Ct value for each primer used was normalised to the corresponding input Ct value: ΔCt normalised m⁶A-IP = Ct m⁶A-IP – [Ct Input –$\log_2$ (Input dilution factor)]. Input dilution factor = (fraction of the input reaction saved)$^{-1}$. As 1% input was saved, the dilution factor is 100 or 6.644 cycles (i.e. $\log_2$ of 100). Percentage of recovery from the initial reaction (% input) was then calculated for each primer as 100 x Amplification efficiency (AE) $^{(-\Delta Ct\ normalised\ m6A-IP)}$. m⁶A enrichment was finally calculated as the fold change between the % input for a region containing m⁶A peaks over the % input for a negative control region. SND1 enrichment was calculated as the fold change between the % input for a region of interest over the % input for a region of a non-target control RNA such as *18S rRNA*. For HEK-293 cells, RIP-qPCR was performed the same way as described for TREx BCBL1-Rta cells with the exception that the sonication time was reduced to eight min and 50 μg of fragmented RNA were used per IP.

## Processing of raw deep-sequencing data and quality control

All m⁶A-seq, RIP-seq and RNA-seq data were generated at the next-generation sequencing facility of the University of Leeds, United Kingdom. Data were extracted and de-multiplexed using bcl2fastq Conversion software (Ilumina), which exports a matched pair (read 1 and read 2) of compressed fastq files per sample. Quality control of all sequence data, including publicly available datasets, was carried out using FastQC software (*Andrews, 2010*), which allowed the identification of sequence adapter contamination, overrepresented sequences, estimation of the PCR and optical duplicate rate and overall sequencing quality. All raw sequence data were then processed using Cutadapt software (*Martin, 2011*) in order to remove poor quality bases (quality score less than 20) as well as Illumina universal sequencing adapter sequence (AGATCGGAAGAG) from the 3' end of reads. Orphan read pairs were discarded. Reads shorter than 25 bp after trimming were discarded to limit ambiguous alignments in downstream processing.

## Next-generation sequencing data alignment and quality control (QC)

The KSHV reference genome sequence was downloaded in FASTA format from the NCBI website, while a gene transfer format (GTF) file containing genomic feature coordinates (ORFs, genes, exons, UTRs) was assembled manually using data from the KSHV 2.0 annotation dataset (*Arias et al., 2014*). The human hg38 reference genome sequence was downloaded from the FTP-UCSC genome browser (*Kent et al., 2002*) in FASTA format. The human hg38 genome annotation was downloaded using the UCSC Table Browser Tool (*Karolchik, 2004*). KSHV data were manually added to the human reference FASTA and GTF files as an additional contig. The genome sequences in the merged FASTA file were indexed for alignment using STAR software (*Dobin et al., 2013*). Paired-end sequence data was subsequently aligned to this index using the splice-aware read aligner STAR, in paired-end, two-pass mode. Aligned reads in binary alignment map (BAM) format were sorted by coordinate and indexed using Samtools (*Li et al., 2009*) and PCR and optical duplicates flagged for additional QC checks (but not removed) using Picard Tools software. Additional QC metrics were assembled and assessed using multiQC software (*Ewels et al., 2016*).

## m⁶A-seq data analysis

m⁶A peaks were called using m6aViewer software version 1.6 (*Antanaviciute et al., 2017*) with default settings and exported to tab-delimited format for additional analyses in R. To define significantly enriched m⁶A peaks in both viral and cellular RNAs, a minimum fold change of m⁶A-IP reads over input reads of ≥1.5 in addition to a false discovery rate of 5% (FDR < 5%) was required in both biological replicates. Peaks positions were considered overlapping between replicates if the calls were within 100 nucleotides between corresponding positions. To determine the number of SND1

RNA targets identified by transcript-wide analysis that are m$^6$A-modified, a more stringent m$^6$A peak calling cut-off was used to compliment a more stringent SND1 RIP cut-off, with a minimum of 100 read paired at the tallest point in the m$^6$A peak and a 2-fold enrichment of m$^6$A-IP reads over input reads using a FDR < 1%. For target overlap between heterologously expressed YTH readers and SND1, high-confidence SND1-bound genes (summarised at HGNC gene symbol annotation level, where multiple Ensembl genes mapped to a symbol, the longest was used) were defined at a cut-off of FDR < 1% and a minimum of 2-fold RIP enrichment over input, while m$^6$A peaks were used as before (FDR < 5%, 1.5 fold minimum enrichment in both replicates).

Peak motif discovery was performed by exporting the flanking 100 base of RNA sequence surrounding peaks in KSHV methylome to a FASTA file using m6aViewer software. Sequences containing repetitive viral sequence were removed. The remaining data was then used for enriched sequence motif detection using the MEME software (*Bailey et al., 2009*), with scrambled sequences used as a control. KSHV methylome maps were produced using custom Java code.

### RIP-seq data analysis

SND1 binding sites were initially identified at transcript-level resolution by counting reads in the SND1 immunoprecipitated (RIP) and input (control) sample data that mapped to each RefSeq and KSHV transcripts. R package *Rsubread* (*Liao et al., 2014*) was used to obtain raw read counts as follows. Each uniquely mapping read pair was counted towards the total transcript count for each sample, while multi-mapping reads were counted as partial reads, based on the number of mapped positions. Since the library preparation protocol preserved the strand of the original RNA molecule, only 'correctly' stranded read pairs were counted for each transcript. DESeq2 R package (*Love et al., 2014*) was used to normalise the data and identify transcripts that showed a significant increase in the coverage of the normalised RIP samples when compared to the input controls. In order to increase the resolution of the SND1-bound regions, custom Java code was used that identified transcriptome regions that were enriched in the RIP data when compared to the control data. Initially, the application segmented regions into intronic or exonic sequences: a region was classified as exonic if the sequence was present in at least one mature transcript. The per-base raw read coverage was determined for both intronic and exonic sequences and normalised using the size factors determined from the earlier normalisation step (DESeq2) to account for library compositions and sizes. Using a sliding window approach, first each mature transcript or intron was divided into contiguous segments that loosely showed putative enrichment in the RIP data (>1 fold enrichment) compared to the input data and those that putatively were depleted in (or equal to) RIP ($\leq$1 fold enrichment). Low coverage segments (<20 reads in RIP) were filtered out as quality control. All data from the different samples in the analysis were then merged to generate a single dataset that contained read depth data for all consensus segments identified this way, both intronic and exonic. Each segmented region was subsequently treated as an individual 'gene' for re-analysis using DESeq2, specifically testing for RIP signal enrichment over input (alternative hypothesis: normalised segment read coverage in RIP greater than in input) in order to identify regions with a significant increase in RIP over input signal.

### Differential expression and functional enrichment analyses

Differential expression analyses were performed in R using DESeq2 package. Functional enrichment analyses were performed using the clusterProfiler R package (*Yu et al., 2012*), using the significance cut-off of <0.05 (Benjamini-Hochberg corrected p-values), with all expressed/detected genes (at least one mapped read pair) used as a background control.

### Splicing analysis

Alternative splicing events were detected using Comprehensive AS Hunting (CASH) (*Wu, 2017*). In addition, the lack of statistically significant splicing events between scramble and SND1-depleted TREx BCBL1-Rta cells highlighted by CASH was also confirmed using Spladder software (*Kahles et al., 2016*) and DEXSeq R package (*Anders et al., 2012*) for detecting differential exon usage (data not shown). Read coverage and splicing graphs were visualised using Integrative Genomics Viewer (IGV) (*Robinson et al., 2011*).

## Publicly available deep-sequencing data

Data downloaded from public repositories were aligned as above, except without the addition of KSHV sequence to the reference genome. The following data were obtained from NCBI's GEO database as raw FASTQ files: HeLa cell line YTHDF1 PAR-CLIP data, two replicates (accessions: GSM1553242, GSM1553243); HeLa cell line YTHDF2 PAR-CLIP data, three replicates (accessions: GSM1197605, GSM1197606, GSM1197607); HeLa cell line YTHDF3 PAR-CLIP data, three replicates (accessions: GSM2424844, GSM2424845, GSM2424846). HepG2 cell line $m^6A$-seq data, four replicates (accessions: GSM2409802, GSM2409803, GSM2715523, GSM2715524). Two replicates of each eCLIP experiment were obtained from the ENCODE database as narrowPeak bed files: HepG2 FXR2 (accessions: ENCFF702QGF, ENCFF638WRZ); HepG2 SND1 (ENCFF471JAQ, ENCFF761CYV); IGF2BP1 (accessions: ENCFF705SDK, ENCFF145YYK); IGF2BP3 (accessions: ENCFF076GHL, ENCFF998WZW); TIAL1 (accessions: ENCFF302ROS, ENCFF467UOO).

## eCLIP analysis

Binding sites from eCLIP experiments were downloaded from ENCODE as narrowPeak bed files. Clusters from replicate one which directly overlapped clusters in replicate two were extracted and kept for downstream analyses. To look at overlaps between eCLIP sites and $m^6A$ peaks, HepG2 RIP-seq data was processed with m6aViewer 1.6 software, using default settings, and overlapping peaks, containing a $\geq$ 1.5 fold increase of $m^6A$-IP reads over input with a FDR < 5% across both replicates were kept. De novo motif analysis of HepG2 $m^6A$-seq and eCLIP sites was performed by feeding peaks into the findMotifsGenome.pl function within the HOMER suite (version 4.9.1) using parameters '-rna -len 5'. For SND1 motif discovery over $m^6A$ exonic regions, exons containing $m^6A$-seq peaks were identified and these whole exon sequences were screened for SND1 peaks. The SND1 peaks within this set of $m^6A$-modified exons were used for motif discovery.

## Supplementary $m^6A$-seq and RIP-seq data

Excel data sheet for all $m^6A$ peaks called in both biological replicates is supplied as *Supplementary file 4*. Excel data sheet for all SND1 targets identified by transcript-wide analysis is supplied as *Supplementary file 5*.

## Data availability

All deep-sequencing data discussed in this publication have been deposited in NCBI's GEO Database, GEO accession number GSE119026. All identified peptides/PSMs for each RNA bait can be found in *Supplementary file 7–15*.

## RNA affinity chromatography and RNA baits sequences

The following biotin-labelled RNA oligonucleotides were centred on the closest GGACU motif to the $m^6A$ peaks detected in *ORF37* and *ORF50* transcripts. For each oligo, one oligo was $m^6A$-modified at the GGACU motif while the control bait remained unmodified. For the ORF37 bait the sequences used were: 5′-biotin-CGGAAAGCUGGCACUGAAGG-$m^6A$-CUUCUUCUAUAGCA UUUCCA-3′ and 5′-biotin-CGGAAAGCUGGCACUGAAGGACUUCUUCUAUAGCAUUUCCA-3′. Baits spanning an $m^6A$ consensus in the first $m^6A$ peak identified in ORF50 (ORF50-1) were: 5′-biotin-UUUGCCAAUCCUGGAGCCAGG-$m^6A$-CUGUUGCCGGCUUCCAUGGUA-3′ and 5′-biotin-UUUGC-CAAUCCUGGAGCCAGGACUGUUGCCGGCUUCCAUGGUA-3′. The sequences for baits spanning an $m^6A$ consensus in the fourth $m^6A$ peak identified in ORF50 (ORF50-4) were: 5′-biotin-GUUG UCCAGUAUUCUGCAAGG-$m^6A$-CUGUACCAGCUGGACACGCCA-3′ and 5′-biotin-GUUGUCCAG UAUUCUGCAAGGACUGUACCAGCUGGACACGCCA-3′. Cropped versions of ORF50-1 (cORF50-1) were: 5′-biotin-GGAGCCAGG-$m^6A$-CUGUUGCCGGCUUC-3′ and 5′-biotin-GGAGCCAGGACUG UUGCCGGCUUC-3′. Stable versions of ORF50-1 (sORF50-1) were: 5′-biotin-UUGGCCCAUCCCG-GAGCCAGG-$m^6A$-CUGUUGCCGGCUUCCGGGGCC-3′ and 5′-biotin-UUGGCCCAUCCCGGAGC-CAGGACUGUUGCCGGCUUCCGGGGCC-3′.

All baits were purchased from Integrated DNA Technologies (IDT).

TREx BCBL1-Rta cells were reactivated with doxycycline for 24 hr followed by lysis of the cells for 25 min in lysis buffer [10 mM NaCl, 2 mM EDTA, 0.5% (v/v) triton X-100, 0.5 mM DTT and 10 mM Tris HCl, pH 7.4] containing complete protease inhibitor cocktail (Roche) and phosphatase inhibitor

cocktail 2 (Sigma-Aldrich). Lysates were centrifuged at 4°C for 10 min at 12,000 x $g$ to pellet cell debris and the supernatant was kept. 1000 µg of protein per pull-down in an approximate volume of 200 µl were supplemented with 40 units of RNasin Plus (Promega) and 50 µg of yeast tRNA (Sigma-Aldrich) and pre-cleared with 30 µl (resin volume) of streptavidin-conjugated agarose beads (Merck Millipore) for 3 hr at 4°C with rotation. The final volume of the binding reaction was topped to 1 mL with binding buffer (150 mM KCl, 1.5 mM $MgCl_2$, 0.05% (v/v) NP-40, 0.5 mM DTT, 10 mM Tris HCl, pH 7.4). While pre-clearing, 30 µl (resin volume) of streptavidin-conjugated agarose beads per pull-down were blocked with 1% (w/v) BSA (Sigma-Aldrich) in PBS (Lonza) and 50 µg of yeast tRNA (Sigma-Aldrich). Pre-cleared lysates were then mixed with the blocked beads and 4 µg of each biotinylated RNA oligo were added per pull-down. Input samples were immediately collected and stored at −80°C until further use. RNA baits were incubated for 2 hr at 4°C with rotation followed by five washes with binding buffer. Proteins were released from the beads by heating at 95°C for 5 min in 30 µl of 2 X Laemmli sample buffer. Input and pull-down samples were used for either Western blotting or LC-MS/MS analysis.

## Mass spectrometry analysis

LC-MS/MS was performed at the proteomics facility of the University of Bristol, United Kingdom. Samples were separated using SDS-PAGE until the dye front had migrated approximately one centimetre into the separating gel. Each gel lane was then excised and subjected to in-gel tryptic digestion using a DigestPro automated digestion unit (Intavis Ltd.). The resulting peptides were fractionated using an Ultimate 3000 nano-LC system in line with an LTQ-Orbitrap Velos mass spectrometer (Thermo Scientific). In brief, peptides in 1% (v/v) formic acid were injected onto an Acclaim PepMap C18 nano-trap column (Thermo Scientific). After washing with 0.5% (v/v) acetonitrile 0.1% (v/v) formic acid, peptides were resolved on a 250 mm ×75 µm Acclaim PepMap C18 reverse phase analytical column (Thermo Scientific) over a 150 min organic gradient, using seven gradient segments (1–6% solvent B over 1 min., 6–15% B over 58 min., 15–32%B over 58 min., 32–40%B over 5 min., 40–90%B over 1 min., held at 90%B for 6 min and then reduced to 1%B over 1 min.) with a flow rate of 300 nl min$^{-1}$. Solvent A was 0.1% formic acid and Solvent B was aqueous 80% acetonitrile in 0.1% formic acid. Peptides were ionised by nano-electrospray ionisation at 2.1 kV using a stainless-steel emitter with an internal diameter of 30 µm (Thermo Scientific) and a capillary temperature of 250°C. Tandem mass spectra were acquired using an LTQ- Orbitrap Velos mass spectrometer controlled by Xcalibur 2.1 software (Thermo Scientific) and operated in data-dependent acquisition mode. The Orbitrap was set to analyse the survey scans at 60,000 resolution (at m/z 400) in the mass range m/z 300 to 2000 and the top twenty multiply charged ions in each duty cycle selected for MS/MS in the LTQ linear ion trap. Charge state filtering, where unassigned precursor ions were not selected for fragmentation, and dynamic exclusion (repeat count, 1; repeat duration, 30 s; exclusion list size, 500) were used. Fragmentation conditions in the LTQ were as follows: normalised collision energy, 40%; activation q, 0.25; activation time 10 ms; and minimum ion selection intensity, 500 counts. The raw data files were processed and quantified using Proteome Discoverer software v1.4 (Thermo Scientific) and searched against the UniProt Human database (downloaded October 2015; 131351 sequences) plus KSHV protein sequences using the SEQUEST algorithm. Peptide precursor mass tolerance was set at 10ppm, and MS/MS tolerance was set at 0.8 Da. Search criteria included carbamidomethylation of cysteine (+57.0214) as a fixed modification and oxidation of methionine (+15.9949) as a variable modification. Searches were performed with full tryptic digestion and a maximum of 1 missed cleavage was allowed. The reverse database search option was enabled and all peptide data was filtered to satisfy false discovery rate (FDR) of 5%.

Comparative reports were produced between methylated and control RNA baits, the data were filtered at 5% FDR and had also removed any proteins that were only matched by a single peptide. In addition, the data was sorted based on the number total number of peptide spectrum matches (PSM's) identified, such that those proteins at the top of the list were only identified in the methylated samples (e.g. no peptides matching that protein were detected in the non-methylated sample). Comparative reports for ORF50-1, ORF50-4 and ORF37 baits are supplied as *Supplementary file 7*, *8* and *9* respectively. To uncover putative m$^6$A readers, all proteins identified for a given methylated and control bait were sorted by total number of PSM's for the m$^6$A bait. Proteins were then classified as enriched in methylated baits if the number of PSM's assigned to the protein was at least

double in the methylated bait compared with the control bait. All identified unique peptides and PSMs for all RNA baits can be accessed on *Supplementary file 10–15*.

Gene-annotation enrichment analysis were performed with The Database for Annotation, Visualisation and Integrated Discovery (DAVID) v6.8.

## RNA secondary structure prediction

RNA secondary structure prediction of baits was carried out using UNAfold web server (*Zuker, 2003*).

## Recombinant protein expression

All recombinant proteins were gene synthesised (NovoPro Bioscience), cloned into pGEX-4T-1 vector (NovoPro Bioscience) and expressed in *E. coli* strain BL21-DE3 (Thermo Scientific). GST-recombinant proteins contained the FXR1 plant agenet domain (residues 2–132) which includes Agenet1 and Agenet2 in tandem, the PSIP1 PWWP domain (residues 3–100) or the CBX3 chromodomain (residues 29–86). SND1-C-terminus comprises of residues 548–910. Recombinant GST plasmid was commercially available (GE healthcare). Recombinant proteins were produced by lowering the temperature to 18°C and inducing the culture with 0.2 mM IPTG for 20 hr. Bacteria pellet from 1 L culture was resuspended with 30 mL of lysis buffer [50 mM Tris HCl, pH 7.5, 150 mM NaCl, 0.05% (v/v) NP-40 and freshly added 0.25 mg/mL lysozyme (Sigma-Aldrich)] and incubated on ice for 45 min. The lysate was then sonicated using a MSE Soniprep 150 sonicator (12 cycles of 20 s pulse-on and 20 s pulse-off). The lysate was centrifuged at 12,000 x *g* for 15 min at 4°C. The supernatant was incubated with glutathione agarose beads (GE healthcare) for 2 hr at 4°C. The resin was washed twice with lysis buffer and once with 50 mM Tris HCl, pH 7.4. Protein was eluted from the beads using 50 mM Tris HCl with 20 mM reduced glutathione (Sigma-Aldrich) that had been adjusted to a final pH of 7.4.

## Electrophoretic mobility shift assays (EMSAs)

EMSAs were carried out using LightShift chemiluminescent RNA EMSA Kit (Thermo Scientific) according to the manufacturer's instructions. Binding reactions consisted of kit supplied 1 X binding buffer (10 mM HEPES, pH 7.3, 20 mM KCl, 1 mM $MgCl_2$ and 1 mM DTT) supplemented with 5% (v/v) glycerol. For EMSAs in the presence of herring sperm DNA (Promega), DNA was mixed and incubated in the binding reaction. Note that this sperm DNA is provided after phenol-chloroform extraction, ethanol precipitation and sonication, which produces single-stranded fragments. Binding reactions were incubated with increasing amounts of recombinant protein which was freshly isolated (e.g. no more than two days after purification from bacteria and stored at 4°C). The same biotinylated baits used in the RNA affinity experiments were used for EMSAs. All baits were used at 3.75 nM oligo final concentration, except for cORF50-1 which was used at 16 nM. Binding reactions were incubated for 30 min at room temperature. 10 µl reactions were mixed with 2 µl of 5 X loading dye and 10 µl were loaded onto a 6% (w/v) non-denaturing polyacrylamide gel in 1 X TAE buffer (40 mM Tris, 20 mM acetate and 1 mM EDTA/NaOH pH 8.0). Gels were ran for 45 min at 100V and transferred to a nylon membrane (GE healthcare) via wet transfer with 1 X TAE as transfer buffer for 45 min at 400 mA. RNA was crosslinked to the membrane at 120mJ/cm$^2$ using a CL-1000 ultraviolet crosslinker (UVP). The rest of the protocol was performed as described in the kit manual. Biotinylated baits were visualised after exposure to an ultra-sensitive ECL (SuperSignal west femto maximum sensitivity substrate) (Thermo Scientific) and exposed to Amersham hyperfilm ECL (GE Healthcare).

## Site-directed mutagenesis

A FLAG tag *ORF50* gene which included both *ORF50* exons and the *ORF50* intron cloned into a pCDH-CMV-MCS-EF1-Puro vector was purchased from NovoPro Bioscience. Single point mutations in this plasmid were generated using QuickChange II site-directed mutagenesis kit (Stratagene) according to the manufacturer's instructions. All plasmids containing the desired mutation were confirmed by DNA sequencing (Eurofins Genomics). To mutate the GGACT motif present in ORF50-1 bait the following primers were used: TCCTGGAGCCAGGATTGTTGCCGG (forward), CCGGCAACAATCCTGGCTCCAGGA (reverse). To mutate the GGACT motif present in ORF50-4 bait these

primers were used: TCCAGTATTCTGCAAGGATTGTACCAGCTGGACAC (forward), GTGTCCAGC
TGGTACAATCCTTGCAGAATACTGGA (reverse).

## shRNA stable cell lines

Lentiviruses were generated by transfection of HEK-293T cells seeded in 12-well plates using a three-plasmid system. Per 12-well, 4 µl of lipofectamine 2000 (Thermo Scientific) were used together with 1 µg of pLKO.1 plasmid expressing shRNA against the protein of interest, 0.65 µg of pVSV.G, and 0.65 µg psPAX2. pVSV.G and psPAX2 were a gift from Dr. Edwin Chen at the University of Leeds. 8 hr post-transfection, medium was changed into 1.5 mL of DMEM supplemented with 10% (v/v) FCS. Two days post-transfection viral supernatants were harvested, filtered through a 0.45 µm filter (Merck Millipore) and immediately used for transductions of TREx BCBL1-Rta or BCBL1 cells. One mL of each 12-well plate was used to infect 500,000 cells by spin inoculation for 60 min at 800 x *g* at room temperature, in the presence of 8 µg/mL of polybrene (Merck Millipore). Virus supernatant was removed after 7 hr post-spin inoculation and cells were maintained in fresh growth medium for 48 hr before undergoing 3 µg/mL puromycin (Sigma-Aldrich) selection. Stable cell lines were generated after 8 days post-selection when cell stocks were frozen. The following shRNAs were used in the experiments: SND1 KD1 (TRCN0000245143), SND1 KD2 (TRCN0000049656), METTL3 KD1 (TRCN0000289742), METTL3 KD2 (TRCN0000289812), FTO KD1 (TRCN0000246247), FTO KD2 (TRCN0000246250), YTHDF1 KD1 (TRCN0000062771), YTHDF1 KD2 (TRCN0000286871), YTHDF2 KD1 (TRCN0000254411), YTHDF2 KD2 (TRCN0000265510), YTHDF3 KD1 (TRCN0000365164) and YTHDF3 KD2 (TRCN0000167772).

All shRNA plasmids were purchased from either Sigma-Aldrich or Dharmacon. Scramble shRNA was a gift from Professor David Sabatini (Addgene plasmid # 1864).

## Western blot analysis

Western blots were performed as previously described (*Schumann et al., 2017*). In brief, protein samples were run on SDS-PAGE gels and transferred onto nitrocellulose membrane (GE healthcare) via wet transfer. Membranes were blocked with TBS + 0.1% (v/v) Tween 20 (TBST) and 5% (w/v) dried skimmed milk powder for 30 min, and then incubated for 1 hr with relevant primary and secondary antibodies diluted in 5% (w/v) milk TBST. Membranes were treated with either ECL Western blotting substrate (Promega) or SuperSignal West Femto Maximum Sensitivity Luminol/Enhancer solution (Thermo scientific) and exposed to Amersham hyperfilm ECL (GE Healthcare). Secondary antibodies were horseradish peroxidase (HRP)-conjugated polyclonal goat anti-mouse (Dako) and polyclonal goat anti-rabbit (Dako), both used at 1:5000 dilution.

## Two-step quantitative reverse transcription PCR (qRT-PCR)

qRT-PCR was performed as previously described (*Baquero-Pérez and Whitehouse, 2015*). In brief, total RNA from cells was extracted using TRIzol (Thermo Scientific) according to the supplier's protocol. DNA-*free* DNA Removal Kit (Ambion) was used to remove any contaminating DNA from RNA samples. Reverse transcription was performed with ProtoScript II (NEB), murine RNase inhibitor (NEB), random hexamers (Bioline) and 1 µg of total RNA. Quantitative PCR (qPCR) reactions (20 µl) included 1 X SensiMix SYBR green master mix (Bioline), 0.5 µM of each primer and 5 µl template cDNA. Cycling was performed in a RotorGene Q 2plex machine (Qiagen). The cycling programme was a 10 min initial preincubation at 95°C, followed by 40 cycles of 95°C for 15 s, 60°C for 30 s and 72°C for 20 s. After qPCR, a melting curve analysis was performed between 65°C and 95°C (with 0.2°C increments) to confirm amplification of a single product. To assess primer amplification efficiency (AE), for each gene of interest a standard curve was constructed using a pool of cDNA derived from unreactivated and reactivated cells. At least four different dilutions of pool cDNA were quantified to generate a standard curve. The slope of the standard curve was used to calculate the AE of the primers using the formula: $AE = (10^{-1/slope})$. For gene expression analysis all genes of interest were normalised against the housekeeping gene *GAPDH* ($\Delta C_T$). A summary of all the primers used in this study is provided in *Supplementary file 3*. Primers specific to METTL3 have previously been described (*Liu et al., 2014*).

## RNA stability assays

TREx BCBL1-Rta cells were treated with 2.5 µg/ml of actinomycin D (Thermo Scientific) and samples were collected at the desired time points. Total RNA was extracted using TRIzol (Thermo Scientific) according to the supplier's protocol. DNA-*free* DNA Removal Kit (Ambion) was used to remove any contaminating DNA from RNA samples and qRT-PCR was carried out as described above. The gene of interest was normalised against *18S rRNA*, as this RNA, but not *GAPDH*, was stable after 6 hr of actinomycin D treatment.

## Chromatin immunoprecipitation (ChIP)

ChIP experiments were carried out as previously described (*Baquero-Pérez and Whitehouse, 2015*). Formaldehyde-crosslinked chromatin was prepared using the Pierce Chromatin Prep Module (Thermo Scientific) following the manufacturer's protocol with the following modified steps. $2 \times 10^6$ BCBL1 cells were used per immunoprecipitation and digested with 15 units of micrococcal nuclease (MNase) per 100 µl of MNase Digestion buffer in a 37°C water bath for 15 min. Nuclei were lysed for 30 min with lysis buffer two in addition to vortexing for 30 s every 5 min. Immunoprecipitations were carried out using EZ-ChIP kit (Millipore) according to the supplier's instructions. Immunoprecipitations were done overnight at 4°C and contained 50 µl of digested chromatin ($2 \times 10^6$ cells), 450 µl of ChIP dilution buffer and 2 µg of the antibody of interest. Primers for the KSHV promoter regions of *ORF50*, *Ori-Lyt*, *PAN RNA*, *ORF59* and *K12* have been previously reported (*Chen et al., 2009b*; *Chen et al., 2012*; *Hughes et al., 2015*). Primers for the cellular *GAPDH* promoter were supplied with the EZ-ChIP kit. qPCR normalisation was performed similarly to RIP-qPCR experiments as described above.

## Proliferation (MTS) assay

Determination of the cellular metabolic activity was performed using a non-radioactive CellTiter 96 AQ$_{ueous}$ One Solution Cell Proliferation Assay (MTS) (Promega), according to the manufacturer's manual. 10,000 HEK-293 cells were seeded in quadruplicate in a flat 96-well culture plate (Corning). After 26 hr inhibitor exposure, 20 µl of CellTiter 96 AQueous One Solution Reagent was added and cells were incubated for 1 hr in a humidified incubator in 5% $CO_2$ at 37°C. Absorbance was measured at 490 nm using a PowerWave XS2 (BioTek) plate reader.

## Acknowledgements

We thank Dr. Kate Heesom (Proteomics facility, University of Bristol, UK) for proteomic technical assistance and initial sorting of proteomic data, Dr. Sally Fairweather and Ummey Hany for technical assistance with library preparation, Dr. Andrew Tuplin (University of Leeds, UK) for help designing sORF50-1 bait, Joseph Snowden (University of Leeds, UK) for technical assistance with EMSAs, Friedrich Grässer (University of Homburg, Germany) for the mouse monoclonal anti-ORF54 and Professor David Sabatini for scramble shRNA. This work was supported by BBRSC (BB/M006557/1) and MRC (MR/M009084/1; MR/R010145/1).

## Additional information

### Funding

| Funder | Grant reference number | Author |
| --- | --- | --- |
| Biotechnology and Biological Sciences Research Council | BB/M006557/1 | Ade Whitehouse |
| Medical Research Council | MR/R010145/1 | Ade Whitehouse |

The funders had no role in study design, data collection and interpretation, or the decision to submit the work for publication.

## Author contributions
Belinda Baquero-Perez, Conceptualization, Resources, Formal analysis, Validation, Investigation, Visualization, Methodology, Writing—original draft, Project administration, Writing—review and editing; Agne Antanaviciute, Data curation, Software, Formal analysis, Validation, Investigation, Visualization, Methodology, Writing—review and editing; Ivaylo D Yonchev, Formal analysis, Investigation, Writing—review and editing; Ian M Carr, Software, Supervision, Project administration; Stuart A Wilson, Formal analysis, Supervision, Project administration, Writing—review and editing; Adrian Whitehouse, Conceptualization, Formal analysis, Supervision, Funding acquisition, Validation, Methodology, Project administration, Writing—review and editing

## Author ORCIDs
Belinda Baquero-Perez (iD) https://orcid.org/0000-0002-0956-7164
Adrian Whitehouse (iD) https://orcid.org/0000-0003-3866-7110

## Decision letter and Author response
Decision letter https://doi.org/10.7554/eLife.47261.056
Author response https://doi.org/10.7554/eLife.47261.057

---

# Additional files

## Supplementary files
• Supplementary file 1. Proteins related to RNA processing are enriched in methylated baits. The number of unique peptides sequences and peptide spectrum matches (PSM's) assigned to each protein as identified by mass spectrometry is displayed for each bait.
DOI: https://doi.org/10.7554/eLife.47261.032

• Supplementary file 2. Proteins with methyl-transferase activity are enriched in methylated baits. While proteins with methyl-transferase activity (highlighted in bold) were recruited to methylated viral baits, neither $m^6A$ indirect readers nor IGF2BP proteins were enriched in any of the viral baits. The number of unique peptides sequences assigned to each protein as identified by mass spectrometry is displayed for each bait.
DOI: https://doi.org/10.7554/eLife.47261.033

• Supplementary file 3. List of all primers used in qPCR experiments.
DOI: https://doi.org/10.7554/eLife.47261.034

• Supplementary file 4. List of cellular $m^6A$ peaks called in latent and lytic TREx BCBL1-Rta cells.
DOI: https://doi.org/10.7554/eLife.47261.035

• Supplementary file 5. List of SND1 RNA targets identified by RIP-seq in TREx BCBL1-Rta cells.
DOI: https://doi.org/10.7554/eLife.47261.036

• Supplementary file 6. List of differential SND1-binding events to target RNAs in TREx BCBL1-Rta cells.
DOI: https://doi.org/10.7554/eLife.47261.037

• Supplementary file 7. Comparative LC-MS/MS report for ORF50-1 baits.
DOI: https://doi.org/10.7554/eLife.47261.038

• Supplementary file 8. Comparative LC-MS/MS report for ORF50-4 baits.
DOI: https://doi.org/10.7554/eLife.47261.039

• Supplementary file 9. Comparative LC-MS/MS report for ORF37 baits.
DOI: https://doi.org/10.7554/eLife.47261.040

• Supplementary file 10. List of proteins identified by LC-MS/MS in A-ORF50-1 bait.
DOI: https://doi.org/10.7554/eLife.47261.041

• Supplementary file 11. List of proteins identified by LC-MS/MS in $m^6A$-ORF50-1 bait.
DOI: https://doi.org/10.7554/eLife.47261.042

• Supplementary file 12. List of proteins identified by LC-MS/MS in A-ORF50-4 bait.
DOI: https://doi.org/10.7554/eLife.47261.043

• Supplementary file 13. List of proteins identified by LC-MS/MS analysis in $m^6A$-ORF50-4 bait.

DOI: https://doi.org/10.7554/eLife.47261.044

- Supplementary file 14. List of proteins identified by LC-MS/MS analysis in A-ORF37 bait.
DOI: https://doi.org/10.7554/eLife.47261.045
- Supplementary file 15. List of proteins identified by LC-MS/MS in m⁶A-ORF37 bait.
DOI: https://doi.org/10.7554/eLife.47261.046
- Transparent reporting form DOI: https://doi.org/10.7554/eLife.47261.047

### Data availability

All deep-sequencing data discussed in this publication have been deposited in NCBI's GEO Database, under GEO accession number GSE119026. All identified peptides/PSMs for each RNA bait can be found in Supplementary files 7-15.

The following dataset was generated:

| Author(s) | Year | Dataset title | Dataset URL | Database and Identifier |
|---|---|---|---|---|
| Baquero-Perez B, Antanaviciute A, Carr I, Whitehouse A | 2018 | m6A-RNA mapping, SND1-RNA binding profile mapping and SND1-depletion in KSHV-infected B-lymphocytes | https://www.ncbi.nlm.nih.gov/geo/query/acc.cgi?acc=GSE119026 | Gene Expression Omnibus, GSE119026 |

The following previously published datasets were used:

| Author(s) | Year | Dataset title | Dataset URL | Database and Identifier |
|---|---|---|---|---|
| Wang X, Zhao BS, Roundtree IA, Lu Z, Han D, He C | 2014 | N6-methyladenosine Modulates Messenger RNA Translation Efficiency | https://www.ncbi.nlm.nih.gov/geo/query/acc.cgi?acc=GSE63591 | Gene Expression Omnibus, GSE63591 |
| zhike lu | 2013 | YTHDF2-PAR-CLIP-rep1 A1 | https://www.ncbi.nlm.nih.gov/geo/query/acc.cgi?acc=GSM1197605 | Gene Expression Omnibus, GSM1197605 |

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
