## [Decision Letter]

Thank you for submitting your article "The Tudor SND1 protein is a m^6^A RNA reader essential for KSHV replication" for consideration by *eLife*. Your article has been reviewed by Päivi Ojala as the Senior Editor, a Reviewing Editor, and three reviewers. The following individual involved in review of your submission has agreed to reveal their identity: Nara Lee (Reviewer #3).

The reviewers have discussed the reviews with one another and the Reviewing Editor has drafted this decision to help you prepare a revised submission.

Summary:

This is an interesting investigation into the landscape of m6A modification of KSHV transcripts and the readers of the modified RNAs. Seven members of the "Royal family" of TUDOR domain containing proteins were identified as new putative m6A readers, including SND1. SND1 associates with the KSHV ORF50 transcript in a manner dependent on the m6A modification and SND1 depletion leads to a global impairment of KSHV gene expression. Identification of members of the 'Royal family' as new m6A-readers greatly increases their epigenetic functions beyond protein methylation.

Essential revisions:

1) Reviewers 1 and 3 suggested the authors to map the SND1 binding sites by CLIP-seq rather than RNA-IP.

2) Reviewer 2 suggested the authors to perform a ChIP-seq analysis of the viral genome to exclude the possibility that SND1 binds and activates the ORF50 promoter.

The full reviews are listed below, in the hope that the comments can help to improve the quality of the paper.

*Reviewer #1:*

More evidence is needed to improve the quality of the manuscript.

Essential revisions:

1) I have several concerns for the gel shift assay and the data interpretation:

(A) The components of the 1x binding buffer of the gel shift assay (subsection “Electrophoretic mobility shift assays (EMSAs)”) looked different from that is stated in the manual of the LightShift Chemiluminescent EMSA Kit from ThermoScientific in terms of salt concentrations. This could affect the secondary structure of the probes used in the assay. Also, it is not clear whether the authors used non-specific Poly (dI•dC) as instructed by the kit. Considering the function of SND1 as a transcriptional factor, does the presence of non-specific DNA affect the RNA binding of SND1 in gel shift assay?

(B) In Figure 2, Figure 3—figure supplement 6 and Figure 3—figure supplement 7, the biotin signals from the probe were always overwhelming compared to the shifted probes, making it hard to calculate the ratio of free probe to bound probe and estimate the binding affinity (Kd). Why is this the case? Any way to optimize the imaging condition? Maybe fluorescent dye labelled probe could help.

2) The authors saved the input sample of the RIP-seq saved before crosslinking the cells. Normally people process the input and RIP samples in parallel until before IP. The authors claimed the reason is that the sonicated input gave lower quality sequencing libraries maybe due to overshearing (subsection “RIP-seq”). I still believe that the input sample should be saved after sonication, otherwise it is hard to estimate artifacts introduced by the sonication step. And "overshearing" itself may already be a concern for high-quality RIP assays. Instead of "formaldehyde-crosslinked RIP-seq", the authors may consider performing "CLIP-seq" as a validation which also gives high-resolution binding sites of protein on RNA based on crosslinking-induced mutations in cDNA libraries.

3) "SND1 binds symmetrically demethylated arginines (sDMA)" via its Tudor domain, as described on subsection “RNA affinity identifies putative m6A readers which belong to the Tudor domain ‘Royal family’”. I am wondering if the authors have tested affinity of the protein towards N6, N6-dimethyladenosine. It may worth checking if known ribosomal RNA N6, N6-dimethyladenosine sites were enriched in their SND1 RIP-seq data.

*Reviewer #2*:

The authors cannot demonstrate the final consequences of SND1 binding to the ORF50 RNA. Here the manuscript reveals conceptual deficits. Also, the work of another laboratory with similar findings is not discussed sufficiently.

Essential revisions:

1) Abstract: The authors provide interesting structural data, but the claim that all identified proteins recognize m6A in a structural-dependent manner is an overstatement (see also below). To verify this more mutants are necessary. The critical claim of the manuscript is that SND1 stabilizes the ORF50 RNA. However, the single data figure given in Figure 6 does not clarify this point (see below). The following sentence mentions a global impairment of KSHV gene expression, which is no surprise if the master switch of reactivation is blocked. This claim has to be toned down to "inhibits KSHV early gene expression".

2) Subsection “Royal domains bind m6A-modified RNA hairpins in a RNA secondary structure-dependent manner”, Figure 2, Figure 3—figure supplement 4 and Figure 3—figure supplement 4: First, these data nicely show that SND1 binds the ORF50-1 sequence. However, the interpretation of the cORF50-1 shortened stem and the sORF50-1 mutant is not straightforward. To state "in structural-dependent manner" the authors need to increase the stability of the stem by erasing the bulges only. Also, the cORF50-1 sequence may be just too short to form the correct stem. The authors should extend the stem of cORF50-1 mutant by unrelated bases to provide more insights on the structural features needed by SND1.

3) Subsection “SND1 is a m6A reader in KSHV-infected cells”, Figure 3: The authors should compare their obtained m6A frequency in the high-confidence SND1 targets to ratios available for other m6A reader proteins such as the YTHDF family members. Is there a similar correlation or do these readers contains even more m6A modified target sequences in their high-confidence interval? These data should be available from public resources.

4) Subsection “SND1 is a m6A reader in KSHV-infected cells”: how SND1 recognizes its RNA target sequences is pure speculation and should be removed from the result section. In addition, the motif provided in Figure 4—figure supplement 3D looks like 3' splice site and may argue for SND1's role in splicing.

5) Subsection “SND1 stabilises *ORF50* RNA and is essential for KSHV replication” and Figure 5: The authors provide an interesting discrepancy between the TREx BCBL-1 cells and naturally infected BCBL-1 cells. Their argument reads as follows: in naturally, unmodified BCBL-1 cells, the ORF50 RNA is decreased during SND1 knockdown. This effect is masked in the TREx BCBL-1 cells since they contain a chromosomally integrated doxycycline-inducible ORF50 gene. However, the authors fail short in the interpretation of their results. An alternative explanation would be that the DNA-binding and promoter-modifying capacity of SND1 comes into play. The major difference between the two ORF50 versions is the nature of the promoter driving their expression. The dox version harbors 6 to 7 tetracyclin-operators and a minimal CMV promoter. In the virus the ORF50 promoter is the sensor of the cellular status and is highly regulated to induced lytic reactivation only under certain conditions. To clarify the role of SND1 the authors need to perform a ChIP-seq analysis of the viral genome to exclude the possibility that SND1 binds and activates the ORF50 promoter. Thus, a lack of SND1 may cause an inhibition of ORF50 transcription and may explain all downstream effects. If this is the case the derived claims such as "SND1 stabilizes the ORF50 transcript" may not be corrected. In addition, the impairment of viral lytic gene expression could also be explained in a similar manner. The authors themselves mention this possibility indirectly in the Discussion section stating that SND1 functions as co-activator of the EBV protein EBNA-2.

6) Figure 5D and G: The authors should include lytic markers such as K8.1 and ORF59.

7) Figure 6B: Here, the authors need to provide the raw data. In addition, to an accelerated decay the SND1 knockdown may already reduce the amount ORF50 transcript. See also argument provided above. Also, the authors should distinguish between the decay of spliced vs. unspliced ORF50 RNA if possible.

8) Supplementary Figure S11: I agree with the authors that a METTL3 knockdown might be difficult in BCBL-1 cells. Even though the increase in ORF50 RNA is not significant I am wondering if the authors here observe the same effect as the Glaunsinger laboratory (Hesser et al., 2018). These authors were able to reduce the amount of METTL3 in BCBL-1 cells sufficiently and observed an increase in ORF50 protein, but not RNA. These findings are not discussed in the current manuscript! If METTL3 knockdown is too difficult, the authors should use the DAA treatment presented in Figure 6C,D to assess ORF50 RNA levels in the absence of m6A modifications.

*Reviewer #3:*

The most exciting part of this manuscript is the identification of novel m6A reader proteins, which were originally thought to bind methylated proteins only. While I am convinced of the specificity of SND1 towards m6A, my main criticism is directed towards the authors' choice of experimental approach to map SND1 binding sites on RNA. As detailed below, RNA-IP in this particular instance appears inappropriate to locate SND1 binding sites.

Essential revisions:

While different experimental approaches can be applied to identify RNA-protein interactions, such as RNA-IP or CLIP-based techniques, in the case of SND1 the use of RNA-IP appears inappropriate. As mentioned in subsection “SND1 is a m6A reader in KSHV-infected cells” and shown in Figure 4—figure supplement 1, the vast majority of RNA fragments was sheared to <200 nt (not "bp"), yet the enriched fragments were of considerably greater size, which the authors suggest may be due to the anti-SND1 antibody having higher affinity for longer fragments. This statement is not very plausible. On the other hand, given the affinity of SND1 for methylated arginines, it is possible that chromatin-associated nascent RNAs were indirectly precipitated through SND1 binding to other chromatin-associated factors (e.g. histones or spliceosomal proteins). These complexed RNAs would greatly resist shearing by sonication as compared to free RNAs. According to Materials and methods section, no DNase treatment was included prior to IP and hence indirect RNA IP cannot be excluded. While bona fide SND1 targets may certainly be present within the deep sequenced RNAs, this cohort may be largely polluted by non-specific transcripts.

Rather than RNA-IP, eCLIP or iCLIP would have the advantage that the crosslinked sites can be identified unambiguously, which would be particularly useful in overlapping m6A sites on RNA with SND1 binding sites.

---

## [Author Response]

Essential revisions:1) Reviewers 1 and 3 suggested the authors to map the SND1 binding sites by CLIP-seq rather than RNA-IP.

To confirm our SND1 RIP analysis, we have analysed SND1 binding profiles using existing eCLIP datasets from the ENCODE consortium. This has allowed us to compare the binding profiles for established m^6^A reader proteins and SND1. Although these datasets are from different cell lines, Figure 6A, shows that there is an extensive overlap of transcripts which contain m^6^A sites, showing that this analysis approach is valid. Therefore, comparing the overlap between transcripts which are both m^6^A-modified and bound by SND1 from RIP-seq and eCLIP datasets, we observe 1,166 transcripts bound by SND1 of which 88% contained m^6^A sites (Figure 6B). In addition, using this eCLIP dataset it allowed the direct comparison of SND1 binding to other established m^6^A readers using existing eCLIP datasets for these respective proteins (Figure 6C-E). Furthermore, we found comparable binding site overlap for SND1 and established readers with m^6^A peaks, whereas a control protein (TIAL1) showed reduced overlap between m^6^A peaks and its binding sites. Importantly, we also utilised HOMER to identify motifs bound by SND1 using the ENCODE eCLIP datasets, which excitingly revealed that SND1 has a high scoring binding motif which matches the core of the m^6^A consensus motif (Figure 6f). We thank the reviewers for this suggestion, which has led to this important new discovery. These results of all the new eCLIP analysis has been incorporated as Figure 6, subsection “SND1 is an m6A reader in KSHV-infected cells” and in the Materials and methods section.

2) Reviewer 2 suggested the authors to perform a ChIP-seq analysis of the viral genome to exclude the possibility that SND1 binds and activates the ORF50 promoter.

We agree with the reviewer that this a critical check, however we consider a genomic-wide analysis unnecessary to address this concern, as our RNA-seq analysis on the SND1 knockdown cell line showed that KSHV replication is basically unchanged in TREx BCBL1-Rta cells in the absence of SND1, with no significant downregulation of any of the KSHV ORFs (Figure 7E) suggesting that SND1 may not be necessary to promote transcription. Thus, the key question is addressing whether SND1 may bind the *ORF50* promoter in BCBL1 cells. Therefore, to conclusively address this query we have performed ChIP coupled to qPCR detection with the control antibody RNA polymerase II (RNAPII), the rabbit SND1 antibody and a non-specific isotype antibody. We tested the occupancy of these proteins at the viral *ORF50* promoter and several other viral promoters in BCBL1 cells. Primers for the cellular *GAPDH* promoter were used as positive control for RNAPII antibody. Whilst RNAPII was present in 4 viral out of 5 viral promoters tested at different levels of occupancy, there was no specific enrichment for either SND1 or the negative control antibody when using 2 µg of antibody per ChIP (the same concentration used for RIP experiments). Note that the rabbit SND1 antibody is a ChIP-grade antibody as demonstrated by our formaldehyde-fixed RIP-seq experiments, in which cells are subjected to identical formaldehyde fixation conditions, and was able to efficiently immunoprecipitate SND1 protein. The same lot number (00020506) of SND1 antibody (Proteintech, 10760-1-AP) was used for western blot, RIP and ChIP experiments. Consequently, we conclude that SND1 does not bind the *ORF50* promoter. These data has been incorporated as Figure 7 —figure supplement 3 and discussed in the manuscript (subsection “SND1 stabilises *ORF50* RNA and is essential for KSHV replication” and Materials and methods section).

The full reviews are listed below, in the hope that the comments can help to improve the quality of the paper.

Reviewer #1:

More evidence is needed to improve the quality of the manuscript.Essential revisions:1) I have several concerns for the gel shift assay and the data interpretation:(A) The components of the 1x binding buffer of the gel shift assay (subsection “Electrophoretic mobility shift assays (EMSAs)”) looked different from that is stated in the manual of the LightShift Chemiluminescent EMSA Kit from ThermoScientific in terms of salt concentrations. This could affect the secondary structure of the probes used in the assay. Also, it is not clear whether the authors used non-specific Poly (dI•dC) as instructed by the kit.

Thermo Scientific offers two different kits to perform EMSA. One kit is optimised to assess DNA/protein interactions LightShift Chemiluminescent EMSA Kit (cat no: 20148) in which the binding buffer composition is: 10X Binding Buffer: 100mM Tris, 500mM KCl, 10mM DTT; pH 7.5.

The kit we used in this manuscript is optimised to assess RNA/protein interactions, the LightShift Chemiluminescent RNA EMSA Kit (cat no: 20158), in which the binding buffer composition is 10X REMSA Binding Buffer: 100mM HEPES (7.3), 200mM KCl, 10mM MgCl2, 10mM DTT.

The use of Poly (dI•dC) is instructed for the DNA/protein interactions kit. We did not alter the RNA EMSA kit binding buffer composition as this has been optimised for RNA/protein interactions. In addition when we isolated SND1-C-terminus protein we noticed that this protein was particularly sensitive to salt concentrations, as when we dialysed the protein after isolation into 10mM HEPES (7.3), 200 mM KCl, 1mM MgCl2, 1 mM DTT and 0.1% triton X (v/v), SND1 precipitated out of solution.

Considering the function of SND1 as a transcriptional factor, does the presence of non-specific DNA affect the RNA binding of SND1 in gel shift assay?

This is an interesting question, therefore, we have repeated the EMSAs in the absence or presence of herring sperm DNA. In the presence of this non-specific DNA the interaction between SND1 and the m^6^A-modified RNA bait (ORF50-1) was abolished, as shown by the disappearance of the shifted bait and full detection of the free ORF50-1 bait. This result is not surprising as SND1 is a known transcription factor and this result suggests that excess DNA may sequester SND1 away from RNA. This experiment has been incorporated to the manuscript as figure 3 —figure supplement 7C and discussed in the manuscript (subsection “Royal domains display selectivity for specific m6A-modified RNA hairpins”).

(B) In Figure 2, Figure 3—figure supplement 6 and Figure 3—figure supplement 7, the biotin signals from the probe were always overwhelming compared to the shifted probes, making it hard to calculate the ratio of free probe to bound probe and estimate the binding affinity (Kd). Why is this the case? Any way to optimize the imaging condition? Maybe fluorescent dye labelled probe could help.

The overwhelming signal for the free probe is due to the fact that very small amount of RNA was shifted in comparison with the free probe (which is consistent with the moderate SND1 affinity for methylated ORF50-1, see Figure 3—figure supplement 2). A shorter (Author response image 1,left image, photo) vs a longer exposure (Author response image 1, right image, scanned) is shown above to better compare the shifted RNA versus the free bait ratio, but even with really short exposures a unique neat band for the free probe was not seen due to exacerbated signal, making impossible to estimate Kd. Because the RNA EMSA kit makes use of a highly sensitive chemiluminescent substrate, the signal from the free probes were always really strong in comparison with the shift. We tried FAM fluorescent dye labelled baits, but due to the lack of amplification of signal, such as it is the case in the biotin/streptavidin/ECL system, this system did not have enough sensitivity to detect a clear shift.

This sentence has been included in the main manuscript (subsection “Royal domains display selectivity for specific m6A-modified RNA hairpins”): “Note that the membranes had to be overexposed to obtain a good shift signal due to the small amount of shifted RNA in comparison with the free bait, consistent with the modest enrichment of SND1 in m^6^A-ORF50-1 bait (Figure 3—figure supplement 2). Similarly, a weak shift has also been previously observed in EMSAs when using FMRP protein (Edens et al., (2019) and IGF2BP proteins (Huang et al., (2018).”

2) The authors saved the input sample of the RIP-seq saved before crosslinking the cells. Normally people process the input and RIP samples in parallel until before IP. The authors claimed the reason is that the sonicated input gave lower quality sequencing libraries maybe due to overshearing (subsection “RIP-seq”). I still believe that the input sample should be saved after sonication, otherwise it is hard to estimate artifacts introduced by the sonication step. And "overshearing" itself may already be a concern for high-quality RIP assays. Instead of "formaldehyde-crosslinked RIP-seq", the authors may consider performing "CLIP-seq" as a validation which also gives high-resolution binding sites of protein on RNA based on crosslinking-induced mutations in cDNA libraries.

See response in essential revisions 1 above. We have removed the sentence “The lower quality libraries may have been a result of overshearing” as we have no evidence to demonstrate this. We do have evidence that after sonication the input RNA was of equivalent size (Figure 4—figure supplement 1A and 1B) to the one generated for m^6^A-seq libraries.

3) "SND1 binds symmetrically demethylated arginines (sDMA)" via its Tudor domain, as described on subsection “RNA affinity identifies putative m6A readers which belong to the Tudor domain ‘Royal family’”. I am wondering if the authors have tested affinity of the protein towards N6, N6-dimethyladenosine. It may worth checking if known ribosomal RNA N6, N6-dimethyladenosine sites were enriched in their SND1 RIP-seq data.

Our SND1 RIP-seq samples were ribosomal depleted before cDNA library construction, thus we are unable to check this. However, it is possible that SND1 may have the ability to bind other RNA-modifications, this has been included in the discussion as follows: “Finally, it is worth pointing out that it exists the possibility that SND1 may be able to read other RNA methyl-modifications in addition to m^6^A. Further studies characterising these methyl-modifications and their corresponding readers will help explore this possibility.” Discussion section.

Reviewer #2:

The authors cannot demonstrate the final consequences of SND1 binding to the ORF50 RNA. Here the manuscript reveals conceptual deficits. Also, the work of another laboratory with similar findings is not discussed sufficiently.Essential revisions:1) Abstract: The authors provide interesting structural data, but the claim that all identified proteins recognize m6A in a structural-dependent manner is an overstatement (see also below). To verify this more mutants are necessary. The critical claim of the manuscript is that SND1 stabilizes the ORF50 RNA. However, the single data figure given in Figure 6 does not clarify this point (see below). The following sentence mentions a global impairment of KSHV gene expression, which is no surprise if the master switch of reactivation is blocked. This claim has to be toned down to "inhibits KSHV early gene expression".

The Abstract has been modified to address these comments.

2) Subsection “Royal domains bind m6A-modified RNA hairpins in a RNA secondary structure-dependent manner”, Figure 2, Figure 3—figure supplement 4 and Figure 3—figure supplement 4: First, these data nicely show that SND1 binds the ORF50-1 sequence. However, the interpretation of the cORF50-1 shortened stem and the sORF50-1 mutant is not straightforward. To state "in structural-dependent manner" the authors need to increase the stability of the stem by erasing the bulges only. Also, the cORF50-1 sequence may be just too short to form the correct stem. The authors should extend the stem of cORF50-1 mutant by unrelated bases to provide more insights on the structural features needed by SND1.

We agree with the reviewer that to state “in a structural-dependent manner” may be an overstatement. However, the proposed experiments do not clarify either how the structure differs between the different RNA baits in the presence or absence of m^6^A. The use of SHAPE or NMR would help addressing this question, but this is beyond the scope of this manuscript. Consequently, we have changed the abstract and the sections in which we stated the royal domains bound “in structural-dependent manner”. For example in the Introduction.

3) Subsection “SND1 is a m6A reader in KSHV-infected cells”, Figure 3: The authors should compare their obtained m6A frequency in the high-confidence SND1 targets to ratios available for other m6A reader proteins such as the YTHDF family members. Is there a similar correlation or do these readers contains even more m6A modified target sequences in their high-confidence interval? These data should be available from public resources.

This information has been added. As stated in the original manuscript, we observe a 50% of SND1 high confidence targets are m^6^A methylated. We have now compared PAR-CLIP and m^6^A-seq datasets and shown that ~65% of total target RNAs bound by YTH readers are m^6^A methylated. See figure 4 —figure supplement 2 and subsection “SND1 is an m6A reader in KSHV-infected cells”.

4) Subsection “SND1 is a m6A reader in KSHV-infected cells”: how SND1 recognizes its RNA target sequences is pure speculation and should be removed from the result section. In addition, the motif provided in Figure 4—figure supplement 3D looks like 3' splice site and may argue for SND1's role in splicing.

This has been removed.

5) Subsection “SND1 stabilises ORF50 RNA and is essential for KSHV replication” and Figure 5: The authors provide an interesting discrepancy between the TREx BCBL-1 cells and naturally infected BCBL-1 cells. Their argument reads as follows: in naturally, unmodified BCBL-1 cells, the ORF50 RNA is decreased during SND1 knockdown. This effect is masked in the TREx BCBL-1 cells since they contain a chromosomally integrated doxycycline-inducible ORF50 gene. However, the authors fail short in the interpretation of their results. An alternative explanation would be that the DNA-binding and promoter-modifying capacity of SND1 comes into play. The major difference between the two ORF50 versions is the nature of the promoter driving their expression. The dox version harbors 6 to 7 tetracyclin-operators and a minimal CMV promoter. In the virus the ORF50 promoter is the sensor of the cellular status and is highly regulated to induced lytic reactivation only under certain conditions. To clarify the role of SND1 the authors need to perform a ChIP-seq analysis of the viral genome to exclude the possibility that SND1 binds and activates the ORF50 promoter. Thus, a lack of SND1 may cause an inhibition of ORF50 transcription and may explain all downstream effects. If this is the case the derived claims such as "SND1 stabilizes the ORF50 transcript" may not be corrected. In addition, the impairment of viral lytic gene expression could also be explained in a similar manner. The authors themselves mention this possibility indirectly in the Discussion section stating that SND1 functions as co-activator of the EBV protein EBNA-2.

See response in essential revision 2 section above.

6) Figure 5D and G: The authors should include lytic markers such as K8.1 and ORF59.

Markers for the lytic ORF54 protein have now been included, see updated Figure 7D and 7G. We have also quantified by qPCR lytic PAN levels in both TREX and BCBL1 cells (see updated Figure 7C and 7F). This has been included in the manuscript subsection “SND1 stabilises *ORF50* RNA and is essential for KSHV replication”.

7) Figure 6B: Here, the authors need to provide the raw data. In addition, to an accelerated decay the SND1 knockdown may already reduce the amount ORF50 transcript. See also argument provided above. Also, the authors should distinguish between the decay of spliced vs. unspliced ORF50 RNA if possible.

The decay of spliced *ORF50* RNA was assessed with specific primers on the same samples used in Figure 8B, the result has been included in the manuscript, Figure 8B. The effect on decay was not as pronounced as for the observed for unspliced *ORF50* RNA, consistent with our SND1 binding results (Figure 8D), in which depletion of m^6^A in *ORF50* RNA particularly abrogated SND1 binding to the unspliced form of *ORF50* transcript.

8) Supplementary Figure S11: I agree with the authors that a METTL3 knockdown might be difficult in BCBL-1 cells. Even though the increase in ORF50 RNA is not significant I am wondering if the authors here observe the same effect as the Glaunsinger laboratory (Hesser et al., 2018). These authors were able to reduce the amount of METTL3 in BCBL-1 cells sufficiently and observed an increase in ORF50 protein, but not RNA. These findings are not discussed in the current manuscript! If METTL3 knockdown is too difficult, the authors should use the DAA treatment presented in Figure 6C,D to assess ORF50 RNA levels in the absence of m6A modifications.

We have now repeated the METTL3 knockdowns in BCBL1 transiently, instead of generating stable depleted cell lines. After five days post-lentiviral transduction cells were reactivated for 24hours and viral and protein levels analysed. In addition, we have also created stable cell lines with two different shRNAs targeting FTO, YTHDF1, YTHDF2 and YTHDF3. Our results are all supportive of a positive role of m^6^A-methylation in the KSHV lytic replication, with reduced ORF50 mRNA levels following METTL3, YTHDF1 or YTHDF3, and increased ORF50 mRNA levels following FTO depletion. Our results are consistent with previous studies performed in the same BCBL1 cell line (Ye et al., 2017).

All these data have been incorporated as Figure 7—figure supplement 4 and Figure 7—figure supplement 5 and in the main manuscript, subsection “Removal of m6A in *ORF50* RNA impairs SND1 binding to *ORF50* RNA”.

Reviewer #3:

The most exciting part of this manuscript is the identification of novel m6A reader proteins, which were originally thought to bind methylated proteins only. While I am convinced of the specificity of SND1 towards m6A, my main criticism is directed towards the authors' choice of experimental approach to map SND1 binding sites on RNA. As detailed below, RNA-IP in this particular instance appears inappropriate to locate SND1 binding sites.

See response in essential revisions 1 above. We have now included new analysis of SND1 eCLIP data which shows binding profiles similar to other m^6^A reader proteins and shows that the m^6^A consensus sequence is a high scoring binding motif for SND1.

Essential revisions:While different experimental approaches can be applied to identify RNA-protein interactions, such as RNA-IP or CLIP-based techniques, in the case of SND1 the use of RNA-IP appears inappropriate. As mentioned in subsection “SND1 is a m6A reader in KSHV-infected cells” and shown in Figure 4—figure supplement 1, the vast majority of RNA fragments was sheared to <200 nt (not "bp"), yet the enriched fragments were of considerably greater size, which the authors suggest may be due to the anti-SND1 antibody having higher affinity for longer fragments. This statement is not very plausible. On the other hand, given the affinity of SND1 for methylated arginines, it is possible that chromatin-associated nascent RNAs were indirectly precipitated through SND1 binding to other chromatin-associated factors (e.g. histones or spliceosomal proteins). These complexed RNAs would greatly resist shearing by sonication as compared to free RNAs. According to Materials and methods, no DNase treatment was included prior to IP and hence indirect RNA IP cannot be excluded. While bona fide SND1 targets may certainly be present within the deep sequenced RNAs, this cohort may be largely polluted by non-specific transcripts.Rather than RNA-IP, eCLIP or iCLIP would have the advantage that the crosslinked sites can be identified unambiguously, which would be particularly useful in overlapping m6A sites on RNA with SND1 binding sites.

As mentioned above we have now included new analysis of SND1 eCLIP data which shows binding profiles similar to other m^6^A reader proteins and shows that the m^6^A consensus sequence is a high scoring binding motif for SND1.

We have additionally removed the sentence “suggesting the antibody had higher affinity for longer fragments”.